# Fair Dataset Distillation via Cross-Group Barycenter Alignment

Mohammad Hossein Moslemi [1 2]   Nima Hosseini Dashtbayaz [1]   Zhimin Mei [1]   Bissan Ghaddar [3 4]   Boyu Wang [1 2]

## Abstract

Dataset Distillation aims to compress a large dataset into a small synthetic one while maintaining predictive performance. We show that as different demographic groups exhibit distinct predictive patterns, the distillation process struggles to simultaneously preserve informative signals for all subgroups, regardless of whether group sizes are mildly or severely imbalanced. Consequently, models trained on distilled data can experience substantial performance drops for certain subgroups, leading to fairness gaps. Crucially, these gaps do not disappear by merely correcting group imbalance, since they stem from fundamental mismatches in subgroup predictive patterns rather than from sample-size disparities alone. We therefore formally analyze the interaction between these two sources of bias and cast the solution as identifying a group-imbalance-agnostic barycenter of the predictive information that induces similar representations across all subgroups. By distilling toward this shared aggregate representation, we show that group fairness concerns can be reduced. Our approach is compatible with existing distillation methods, and empirical results show that it substantially reduces bias introduced by dataset distillation. Code is available at https://github.com/mhmoslemi/COBRA

## 1. Introduction

Dataset Distillation (DD) aims to compress a large training set into a much smaller synthetic dataset, while ensuring that models trained on this distilled set reach performance levels comparable to those trained on the original data (Wang et al., 2018). However, as distilled datasets are increasingly deployed in high-stakes applications (e.g., healthcare and finance), matching aggregate accuracy alone is insufficient; the distilled data must also preserve predictive relationships across demographic groups to avoid degraded performance for minority populations. Because distillation aggregates many real samples into a compact synthetic set, it can blur the underlying demographic subgroup structure and undermine the application of training-time fairness constraints that rely on group labels. This creates a need to explicitly design the distilled dataset to be fair by construction.

To explain how these disparities arise, dataset distillation is usually performed by aligning a representation of the real dataset with that of a synthetic one, and then optimizing the synthetic samples to minimize the distance between the two representations (Zhao et al., 2021; Zhao & Bilen, 2023). Yet demographic groups can follow different representation patterns and may be unevenly represented in the original data. Current distillation approaches match the synthetic representation to a single aggregate representation computed over all samples. When groups are imbalanced and their representations differ, this aggregation is dominated by the majority group, causing the distilled data to primarily capture majority group patterns and under-represent minorities. Consequently, models trained on the distilled set can perform worse on some demographic groups than on others. Figure 1a illustrates this effect when groups vary in both representation and size. In fairness-unaware distillation (Vanilla DD), the overall representation ($\times$-mark) drifts toward the majority groups. This issue can be partially addressed by performing uniform sampling over demographic groups (Uniform DD). However, there may still be cases where certain groups are located very close to each other in the representation space. In such situations, their combined representation may match them especially well, drawing the distilled data toward these groups and consequently reducing performance for the other groups, as shown in Figure 1b

In this work, we establish theoretically and validate empirically that under dataset distillation, any bias present in the original data becomes amplified, and this amplification cannot be attributed solely to group imbalance or differences in subgroup representations; rather, it arises from the interaction between these two factors. We additionally propose an upper bound for the interaction between these two. Figure 2 illustrates how each effect contributes. In particular, Figure 2a isolates the role of varying group imbalance

[1]Western University, London, Ontario [2]Vector Institute, Toronto, Ontario [3]Ivey Business School, London, Ontario [4]IE University, Segovia, Spain. Correspondence to: Boyu Wang <bwang@csd.uwo.ca>.

*Proceedings of the 43$^{rd}$ International Conference on Machine Learning*, Seoul, South Korea. PMLR 306, 2026. Copyright 2026 by the author(s).

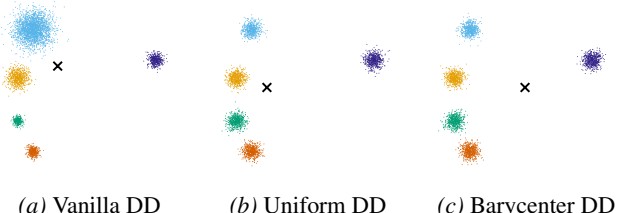

*(a)* Vanilla DD     *(b)* Uniform DD     *(c)* Barycenter DD

*Figure 1.* Representation target used by dataset distillation for different objectives. Colored clouds indicate subgroup representations, and "×" denotes the overall aggregate target.

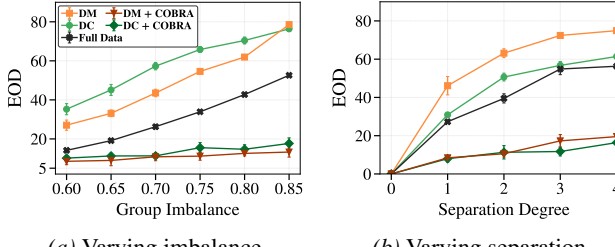

*(a)* Varying imbalance     *(b)* Varying separation

*Figure 2.* **Bias amplification through distillation.** Equalized odds difference (EOD ↓) on CIFAR10-S at IPC = 50 when varying the group-imbalance (2a) with fixed representational separation, and subgroup representational separation (2b) with fixed imbalance. Black curves correspond to full-data training; colored curves to training on distilled sets. A separation level of 0 indicates identical subgroup representations (see Appendix F for details).

while keeping subgroup representations fixed, whereas Figure 2b analyzes the impact of increasing representational separation between subgroups while maintaining a constant imbalance ratio. Both figures indicate that each factor, on its own, has a distinct influence on the outcome.

To tackle this issue, we introduce **COBRA** (**C**ross-gr**O**up **BaR**ycenter **A**lignment), which frames fair distillation as a two-step process: first, we compute a barycenter of the representations of different groups in a manner that is agnostic to group imbalance, and then we distill synthetic data toward this barycenter. A barycenter is a type of average that minimizes the total distance, under a selected discrepancy measure, to a given set of points or distributions (Figure 1c). As a result, this aggregate remains similarly distant from each subgroup, independent of the subgroup's size. Consequently, the distilled dataset captures a comparable level of information and representation quality for all subgroups, thereby mitigating bias. We further demonstrate that COBRA reduces our proposed upper bound on bias amplification by controlling the interaction between group imbalance and representational separation. Consistent with this theory, COBRA achieves state-of-the-art fairness performance relative to vanilla DD and existing baselines. In particular, we make the following contributions:

1. We show that bias amplification in dataset distillation stems from the *interaction* between group imbalance and representational separation across subgroups, and we derive an upper bound formalizing this effect.

2. We introduce a principled strategy to tighten this upper bound, yielding **COBRA**, a framework compatible with any existing dataset distillation method.

3. Comprehensive experiments confirm COBRA's effectiveness, showing consistent fairness gains across diverse datasets and distillation techniques.

**Conflict of Interest Disclosure.** The authors declare no financial or other substantive conflicts of interest that could reasonably be perceived to influence the results reported in this work.

## 2. Related Work

Coresets are small weighted subsets of the training data that approximate learning on the full dataset, with classical constructions for clustering (Jubran et al., 2020; Cohen-Addad et al., 2022; Har-Peled & Mazumdar, 2004), regression (Meyer et al., 2022; Maalouf et al., 2019), and mixture models (Lucic et al., 2018). However, though extending these ideas to deep learning remains challenging, even with gradient matching approaches that use weighted subsets (Mirzasoleiman et al., 2020; Killamsetty et al., 2021; Tukan et al., 2023). Dataset distillation is related to coreset construction, but instead of selecting real samples, it learns *synthetic* samples that mimic the full dataset's training dynamics. It was first posed as a bi-level optimization in (Wang et al., 2018), but direct solutions are computationally costly. This has motivated efficient methods such as kernel based (KIP) (Nguyen et al., 2021), gradient matching (DC, DSA, IDC) (Zhao et al., 2021; Zhao & Bilen, 2021; Kim et al., 2022), feature alignment (CAFE) (Wang et al., 2022), distribution matching (DM) (Zhao & Bilen, 2023), and trajectory matching (Cazenavette et al., 2022; Liu et al., 2024; Yang et al., 2024), along with combinations of those (Zhao et al., 2023; Son et al., 2024; Zhang et al., 2024). In addition to these methods, approaches inspired by domain adaptation have been introduced to tackle distillation on larger scale datasets (D3S) (Loo et al., 2024).

Group fairness seeks comparable outcomes across demographic groups and is crucial in high-stakes domains such as healthcare and finance (Dwork et al., 2012; Hardt et al., 2016). Disparities can originate from demographic imbalance that skews learning toward majority groups (Caton & Haas, 2024) and from conflicting group-dependent predictive mechanisms (Shui et al., 2022). Bias mitigation approaches include pre-processing, in-processing, and post-processing (Louizos et al., 2015; Jung et al., 2022; 2021; Wang et al., 2020). As one pre-processing approach, barycenters and optimal transport have been used to modify

the initial data in order to reduce bias (Gordaliza et al., 2019; Charpentier et al., 2023; Pirhadi et al., 2024).

Fairness in dataset distillation has so far received limited attention. Most related work focuses on class imbalance and long-tailed label distributions (Cui et al., 2024b; Lu et al., 2024; Zhao et al., 2025; Cui et al., 2024a). Additionally, (Vahidian et al., 2025) address out-of-distribution challenges in dataset distillation, but their method is restricted to class labels and does not handle sensitive attributes or group fairness. To the best of our knowledge, (Zhou et al., 2025) is the only existing approach that explicitly aims at subgroup fairness in dataset distillation. They frame the problem as one of group imbalance, compute a separate loss for each subgroup, and then average these losses. While this can mitigate some effects of group imbalance, bias is not determined solely by relative group sizes, and the proposed methods should not be viewed as addressing only this issue. In addition, their method works directly in loss space by averaging per-group losses that have already been computed, rather than first specifying a unified objective across groups and optimizing a single distillation loss. As a result, parameter updates can still differ between groups, limiting the overall effectiveness of the approach. For a more comprehensive discussion of related work, including fairness-aware coresets and data generation, we refer to Appendix B.

# 3. Preliminaries

## 3.1. Dataset Distillation

Let $\mathcal{D}$ be a distribution over $\mathcal{X} \times \mathcal{Y}$, where $\mathcal{X} \subset \mathbb{R}^d$ and $\mathcal{Y} = \{1, \ldots, C\}$. Let $f_\theta : \mathcal{X} \to \mathcal{Y}$ be a predictor with parameters $\theta$, and $\ell(\cdot, \cdot)$ a task loss. For any dataset $U$, define the empirical risk $\mathcal{L}_U(\theta) = \frac{1}{|U|} \sum_{(x,y) \in U} \ell(f_\theta(x), y)$, and let $\theta_U \in \arg\min_\theta \mathcal{L}_U(\theta)$ denote an ERM solution, and write $f_{\theta_U}$ for the resulting predictor. Given a dataset $T = \{(x_i, y_i)\}_{i=1}^{|T|}$ drawn from $\mathcal{D}$, dataset distillation (DD) seeks a much smaller synthetic set $S$ with $|S| \ll |T|$ such that $f_{\theta_S}$ performs comparably to $f_{\theta_T}$ on $\mathcal{D}$, i.e.,

$$\mathbb{E}_{(x,y) \sim \mathcal{D}}[\ell(f_{\theta_T}(x), y)] \simeq \mathbb{E}_{(x,y) \sim \mathcal{D}}[\ell(f_{\theta_S}(x), y)].$$

A standard formulation casts this as a bi-level optimization problem (Wang et al., 2018):

$$S^* = \arg\min_S \mathcal{L}_T(\theta_S) \quad \text{s.t.} \quad \theta_S = \arg\min_\theta \mathcal{L}_S(\theta). \quad (1)$$

Directly solving (1) is computationally expensive. A common practical alternative is to optimize *surrogate objectives* by matching *training signals* (or statistics) induced by $T$ and $S$ under the same learning dynamics (Zhao et al., 2021). The key intuition is that, for a locally smooth predictor, if training on $S$ leads to parameters or representations close to those obtained by training on $T$, then the resulting predictors should behave similarly on real data (and generalize

similarly) (Zhao et al., 2021). Concretely, let $\phi(x; \theta)$ denote a representation of input $x$ from a network with parameters $\theta$. Depending on the method, $\phi$ may correspond to gradients (DC) (Zhao et al., 2021), embeddings (DM) (Zhao & Bilen, 2023), intermediate features (CAFE) (Wang et al., 2022), or training trajectories (MTT) (Cazenavette et al., 2022).

Let $U_y := \{(x, y') \in U : y' = y\}$ be the samples in $U$ with label $y$, and define the class-conditional target statistic

$$\Phi_{U_y} := \frac{1}{|U_y|} \sum_{x \in U_y} \phi(x; \theta_U).$$

Surrogate objectives optimize $S$ by matching these statistics of $T$ and $S$ across classes. Specifically, let $D(\cdot, \cdot)$ be a distance function (e.g., MSE, cosine similarity, MMD, or the $\ell_1$ norm). Then $S^*$ minimizes the following loss:

$$\mathcal{L}(T, S) := \sum_{y \in \mathcal{Y}} D(\Phi_{T_y}, \Phi_{S_y}). \quad (2)$$

## 3.2. Group Fairness

Our goal is to design a model-agnostic algorithm that improves *group fairness*. We study group fairness using the notion of *equalized odds* (EO), which requires conditional independence $A \perp\!\!\!\perp \hat{Y} \mid Y$. Here $\mathcal{A} = \{a_1, a_2, \ldots, a_K\}$ denotes the set of demographic groups, and $A$ is the protected attribute. A sample $(x, y)$ belongs to group $a_i$ if $A = a_i$. Let $\hat{Y} = f(X)$ represent the model's prediction and define $p_y^{\text{EO}}(a) := \Pr(\hat{Y} = y \mid Y = y, A = a)$. Following (Jung et al., 2021), we quantify violations of equalized odds using the maximum EO gap:

$$\text{EOD} = \sup_{y \in \mathcal{Y}, a_i, a_j \in \mathcal{A}} \left| p_y^{\text{EO}}(a_i) - p_y^{\text{EO}}(a_j) \right|. \quad (3)$$

In dataset distillation, we cannot impose constraints on EOD directly because the downstream model is not available. Instead, we perform a label- and group-aware refinement when generating the distilled data. The goal is to better align the class-conditional information across different groups, so that a broad variety of downstream models trained on this fixed dataset yield smaller EO disparities under the above metrics.

# 4. Fair Dataset Distillation

## 4.1. Bias Mechanisms in Dataset Distillation

We characterize how bias arises in distilled data by explicitly explaining how it enters the distillation objective in (2). Given a dataset $T$, a label $y$, and a subgroup attribute $a \in \mathcal{A}$, we define $T_{a|y} := \{(x, y) \in T_y : A(x) = a\}$ and the corresponding class-conditional subgroup proportion $\pi_{a|y} := \frac{|T_{a|y}|}{|T_y|}$. For each subgroup, we define the class-

conditional statistic

$$\Phi_{T_{a|y}} := \frac{1}{|T_{a|y}|} \sum_{x \in T_{a|y}} \phi(x; \theta_T).$$

Substituting these quantities into (2), for each $y \in \mathcal{Y}$:

$$\mathcal{L}_y(T, S) := D\left(\Phi_{T_y}, \Phi_{S_y}\right) = D\left(\sum_{a \in \mathcal{A}} \pi_{a|y}\,\Phi_{T_{a|y}},\ \Phi_{S_y}\right).$$

Let $D$ be the MSE, $D(u, v) = \frac{1}{2}\|u - v\|_2^2$. Then, for each $y \in \mathcal{Y}$, the SGD update of $\Phi_{S_y}$ with step size $\eta_t$ is

$$\nabla_{\Phi_{S_y}} \mathcal{L}_y(T, S) = \Phi_{S_y} - \sum_{a \in \mathcal{A}} \pi_{a|y}\,\Phi_{T_{a|y}},$$

$$\Rightarrow \quad \Phi_{S_y}^{(t+1)} = \Phi_{S_y}^{(t)} - \eta_t \nabla_{\Phi_{S_y}} \mathcal{L}_y(T, S)$$
$$= (1 - \eta_t)\Phi_{S_y}^{(t)} + \eta_t \sum_{a \in \mathcal{A}} \pi_{a|y}\,\Phi_{T_{a|y}}.$$

This implies that $\Phi_{S_y}$ is iteratively pulled toward the class-conditional mixture $\sum_{a \in \mathcal{A}} \pi_{a|y}\Phi_{T_{a|y}}$, and at convergence

$$\Phi_{S_y}^{\star} = \sum_{a \in \mathcal{A}} \pi_{a|y}\,\Phi_{T_{a|y}}, \tag{4}$$

When the collection $\{\Phi_{T_{a|y}}\}_{a \in \mathcal{A}}$ is misaligned and the subgroup-conditional statistics are separated in representation space, the distilled statistic $\Phi_{S_y}$ cannot match all of them at once. We therefore define the subgroup-specific residual as

$$\Delta_{a|y}^{\star} := \Phi_{T_{a|y}} - \Phi_{S_y}^{\star}. \tag{5}$$

Since EOD measures prediction discrepancies across subgroups *conditional on the true label* $y$, and the distillation objective drives $\Phi_{S_y}$ toward the mixture in (4), any predictor subsequently trained on $S$ is implicitly encouraged to fit class-conditional structure centered at $\Phi_{S_y}^{\star}$. When the residuals $\Delta_{a|y}^{\star}$ differ across $a$, the conditional score distributions for distinct subgroups are shifted in different directions, which induces subgroup-specific conditional error patterns and thereby worsens EOD. Intuitively, a larger $\|\Delta_{a|y}^{\star}\|_2$ signals that subgroup $a$ is inadequately represented for label $y$, increasing its conditional error under a model learned from $S$. More formally, by expanding $\Delta_{a|y}^{\star}$, we obtain

$$\Delta_{a|y}^{\star} = \sum_{a' \neq a} \pi_{a'|y}\left(\Phi_{T_{a|y}} - \Phi_{T_{a'|y}}\right),$$

and consequently,

$$\|\Delta_{a|y}^{\star}\|_2 \leq \sum_{a' \neq a} \pi_{a'|y}\,\|\Phi_{T_{a|y}} - \Phi_{T_{a'|y}}\|_2. \tag{6}$$

This indicates that the residual becomes large due to the interaction of two effects: (i) severe subgroup imbalance and (ii) a large separation between subgroup statistics in the representation space. In either scenario, $\Delta_{a|y}^{\star}$ can be large for certain label–subgroup combinations, which degrades conditional accuracy and thus harms EOD.

## 4.2. Subgroup Barycentric Distillation

We previously made the failure mode of distillation for fairness explicit in Equation (6). Motivated by this bound, we introduce a simple adjustment to the class-conditional target that seeks to reduce the impact of both group imbalance and representation separation. We replace the target $\Phi_{T_y}$ in (2) with the *barycenter* of the subgroup-specific representations, where we formulate the barycenter as

$$m_y^{\star} = \arg\min_m \sum_{a \in \mathcal{A}} w_a\, d\left(\Phi_{T_{a|y}}, m\right), \ \forall y \in \mathcal{Y}. \tag{7}$$

Here, $d$ determines the notion of barycenter (e.g., cosine distance or MSE), and $\{w_a\}_{a \in \mathcal{A}}$ represents the importance weights. We select uniform weights $w_a$ to eliminate dependence on $\pi_{a|y}$, thereby preventing majority subgroups from dominating the target. Furthermore, because $m_y^{\star}$ minimizes the total deviation from all subgroup statistics, it serves as the most central compromise when the $\{\Phi_{T_{a|y}}\}$ are dispersed, which contracts differences in $\|\Delta_{a|y}\|_2$ and thereby reduces subgroup-specific error gaps. We now formalize *COBRA*, our proposed fair distillation objective.

**COBRA: Cross-group Barycenter Alignment.** We instantiate subgroup barycentric distillation via the objective

$$\mathcal{L}_{\text{COBRA}}(T, S) := \sum_{y \in \mathcal{Y}} D\left(m_y^{\star},\ \Phi_{S_y}\right) \tag{8}$$

$$\text{s.t.} \quad m_y^{\star} \in \arg\min_m \sum_{a \in \mathcal{A}} d\left(\Phi_{T_{a|y}}, m\right), \quad \forall y \in \mathcal{Y}.$$

For each class $y$, we first compute a cross-group aggregated target representation $m_y^{\star}$ as the barycenter of the subgroup-specific statistics $\{\Phi_{T_{a|y}}\}_{a \in \mathcal{A}}$. We then align the distilled class-conditional representation $\Phi_{S_y}$ with this barycentric target. Under the same assumptions as in Section 4.1, the COBRA objective attains its minimum at $\Phi_{S_y}^{\star} = m_y^{\star}$, meaning that the distilled data matches the barycentric target representation. We now present a theorem showing that the objective $\mathcal{L}_{\text{COBRA}}(T, S)$ decreases the residuals in (5).

**Theorem 4.1.** *Let $d(u, v) = \|u - v\|_Q^2$ for some positive definite matrix $Q \succ 0$, and fix an arbitrary $y \in \mathcal{Y}$. Denote by $m_y^{\star}$ the COBRA barycenter defined in (8) and consider $m_y^{\text{van}} := \sum_{a \in \mathcal{A}} \pi_{a|y}\Phi_{T_{a|y}}$ as the vanilla target from (4). Define the class-conditional residuals*

$$\Delta_{a|y}^{\text{C}} := \Phi_{T_{a|y}} - m_y^{\star}, \qquad \Delta_{a|y}^{\text{V}} := \Phi_{T_{a|y}} - m_y^{\text{van}}.$$

*Assume there exists $a^{\dagger} \in \arg\max_{a \in \mathcal{A}} \|\Delta_{a|y}^{\text{C}}\|_Q$ such that, letting $s_y := m_y^{\text{van}} - m_y^{\star}$,*

$$\langle \Delta_{a^{\dagger}|y}^{\text{C}},\, s_y \rangle_Q \leq 0, \qquad \text{where } \langle u, v \rangle_Q := u^{\top}Qv. \tag{9}$$

*Then COBRA does not increase the worst-case residual:*

$$\max_{a \in \mathcal{A}} \|\Delta_{a|y}^{\text{C}}\|_Q \leq \max_{a \in \mathcal{A}} \|\Delta_{a|y}^{\text{V}}\|_Q. \tag{10}$$

See Appendix C.3 for the detailed proof.

Condition (9) requires that the subgroup achieving the CO-BRA worst-case residual must lie on a non-positive $Q$-angle with the "imbalance shift" $s_y = m_y^{\text{van}} - m_y^\star$. In other words, under vanilla, the target is pulled toward certain subgroups and pushed away from the worst-case one. In this setting, COBRA is guaranteed to reduce the worst-case label-conditional subgroup mismatch in the representation space relative to vanilla. Because EOD captures label-conditional disparities across subgroups, this contraction directly mitigates the geometric mechanism identified in Sec. 4.1 as causing EOD degradation.

## 5. Results

### 5.1. Fairness Aware Dataset Distillation

**Datasets** We evaluate COBRA on biased classification benchmarks that cover both controlled settings and real-world biases. Our controlled suite includes Colored-MNIST (C-MNIST) and Colored-Fashion-MNIST (C-FMNIST), each evaluated under foreground and background spurious correlations, as well as CIFAR10-S with a combined spurious cue. For C-MNIST and C-FMNIST, we create a spurious color attribute by tinting either the foreground object or the background using one of ten fixed colors. For CIFAR10-S, we define a binary spurious attribute by converting images to grayscale or color, and we correlate this attribute with the class label using a fixed group imbalance ratio. The test split is balanced across groups. We also evaluate on real-world datasets with annotated protected attributes, namely UTKFACE and BFFHQ. We additionally report a FULL baseline trained on the complete training set without distillation. For brevity, we refer to group imbalance as SKEW. Further information about the datasets is provided in Appendix E.

**Setup** Across all benchmarks, we distill a compact labeled set $S$ under a fixed images-per-class (IPC) budget. A new evaluation network is then trained from scratch on $S$ and evaluated on the standard test split. We focus on methods that form the backbone of many subsequent approaches and represent distinct architectural paradigms for dataset distillation. In particular, we compare against representative baselines spanning gradient matching (DC, IDC) (Zhao et al., 2021; Kim et al., 2022), distribution matching (DM) (Zhao & Bilen, 2023), and feature alignment (CAFE) (Wang et al., 2022), using their official implementations. We quantify fairness using the metric introduced in Section 3.2, denoted as $\text{EOD}(\downarrow) \in [0, 100]$. Correspondingly, we report accuracy as $\text{Acc}(\uparrow) \in [0, 100]$. We compare our results with those reported in prior work (Zhou et al., 2025), as well as with Vanilla DDs and the FULL scenario. All reported numbers are the averages over 10 independent runs; we only

report the means here, while the corresponding standard deviations are provided in Appendix G.1.

**Choice of Barycenter Discrepancy** Optimizing the CO-BRA objective in (8) is computationally demanding, as it requires repeatedly solving for the barycenter $m_y^\star$ throughout the training of $S$ (and after each surrogate re-initialization). To make this step more tractable, we specialize $d$ to be a squared $Q$-norm with $Q \succ 0$, i.e., $d(u, v) = \|u - v\|_Q^2$, so that the inner barycenter admits a closed-form expression and yields numerically stable solutions. Under this choice, the inner optimization problem has a unique minimizer given by the uniform mean over subgroups, $m_y^\star = \frac{1}{|\mathcal{A}|} \sum_{a \in \mathcal{A}} \Phi_{T_{a|y}}$, which corresponds to the subgroup-wise average representation. A detailed justification of this result is provided in Appendix C.2. Furthermore, we explore alternative definitions of $d$ and their empirical impact in an ablation study presented in Section 5.5.

**Discussion.** Table 1 shows that incorporating COBRA into Vanilla DD objectives reliably yields distilled datasets that depend less on protected attributes at test time and often attain higher predictive performance than both Vanilla DD and its FairDD-augmented variant under the same backbone objective. Across controlled spurious-correlation benchmarks, COBRA consistently reduces large gaps in EOD while preserving or enhancing overall Acc. For example, on C-MNIST (BG) with DM at IPC $= 50$, COBRA lowers EOD from 100.0 to 7.46 and raises accuracy from 48.8 to 96.8. A similar pattern appears on CIFAR10-S with DM at IPC $= 100$, where EOD drops from 82.87 to 9.37 and accuracy increases from 45.4 to 62.4. These fairness gains also persist on real-world data. On BFFHQ with DM at IPC $= 100$, COBRA reduces EOD from 63.47 to 7.87 while improving accuracy from 65.8 to 74.2, indicating that the distilled dataset is both more group-fair and more predictive on the balanced test split. An important implication of these findings is that Vanilla DD can *amplify* existing biases in original datasets. As IPC decreases, this phenomenon intensifies because there is a reduced capacity to represent minority groups and less opportunity to encode within-class diversity. As a result, the distilled dataset can over-represent majority-group evidence and encode stronger spurious correlations than the full dataset, thereby enlarging inter-group performance gaps and yielding large EOD values. Standard deviations for every entry in Table 1, computed under the same 10-run protocol, are reported in Tables 8 and 9 of Appendix G.1.

### 5.2. Group Imbalance & Group Separation Effect

To disentangle the effects of *group imbalance* and *group separation* in representation space (denoted as GAP), we run two controlled experiment sets. First, we hold GAP constant and vary SKEW from 0.60 to 0.85. Second, we fix SKEW

*Table 1.* Fairness-aware dataset distillation on biased benchmarks. Each cell reports EOD ↓ (Acc ↑), averaged over 10 runs. **Bold** highlights the fairness variant achieving the lowest EOD (best fairness) under the same distillation objective for a given dataset and IPC.

| Dataset | IPC | DC | | | DM | | | CAFE | | | IDC | | | FULL |
|---|---|---|---|---|---|---|---|---|---|---|---|---|---|---|
| | | Vanilla | FairDD | COBRA | Vanilla | FairDD | COBRA | Vanilla | FairDD | COBRA | Vanilla | FairDD | COBRA | |
| CIFAR10-S | 10 | 47.12(36.7) | 26.30(40.7) | **17.83**(40.9) | 56.25(37.2) | 25.58(43.9) | **20.18**(44.5) | 62.27(35.8) | 39.97(36.0) | **30.48**(39.5) | 54.60(34.6) | 28.14(40.1) | **22.42**(40.8) | 48.96(69.71) |
| | 50 | 71.85(39.5) | 35.65(46.2) | **26.18**(46.6) | 73.22(45.3) | 25.40(57.7) | **16.70**(57.9) | 61.93(43.5) | 36.10(44.5) | **30.28**(54.1) | 63.54(39.4) | 37.68(51.2) | **25.12**(50.1) | |
| | 100 | 69.37(41.8) | 41.43(49.2) | **21.15**(50.5) | 82.87(45.4) | 25.17(61.2) | **9.37**(62.4) | 73.33(43.2) | 27.73(44.1) | **18.70**(58.0) | 60.64(40.1) | 30.80(51.1) | **22.44**(50.8) | |
| C-FMNIST (FG) | 10 | 99.00(62.0) | 53.83(75.2) | **37.00**(77.0) | 100.0(33.9) | 29.50(76.6) | **25.83**(77.1) | 100.0(27.9) | 45.50(73.7) | **25.50**(74.6) | 100.0(41.8) | 47.20(75.8) | **36.00**(75.9) | 78.40(82.96) |
| | 50 | 100.0(65.7) | 47.50(75.3) | **23.00**(74.5) | 100.0(49.9) | 25.67(81.4) | **21.00**(82.4) | 100.0(48.1) | 50.30(80.3) | **25.40**(81.1) | 100.0(57.0) | 35.00(77.1) | **29.00**(79.5) | |
| | 100 | 100.0(61.2) | 74.83(68.6) | **36.83**(77.2) | 100.0(58.1) | 26.33(82.8) | **24.17**(84.5) | 100.0(54.7) | 21.83(82.0) | **19.50**(83.4) | 100.0(58.0) | 40.60(78.2) | **31.80**(77.4) | |
| C-FMNIST (BG) | 10 | 100.0(49.3) | 61.00(69.0) | **34.50**(70.4) | 100.0(22.1) | 44.67(70.9) | **33.80**(70.6) | 100.0(23.1) | 90.17(59.4) | **46.70**(63.5) | 100.0(33.5) | 47.20(67.8) | **30.40**(67.5) | 90.60(77.84) |
| | 50 | 100.0(61.6) | 50.00(76.1) | **27.17**(73.7) | 100.0(36.7) | 25.17(78.5) | **21.40**(79.8) | 100.0(37.0) | 90.00(66.4) | **53.00**(70.6) | 100.0(41.1) | 36.60(72.6) | **34.00**(71.7) | |
| | 100 | 99.00(63.6) | 50.67(70.7) | **36.67**(75.0) | 100.0(46.2) | 28.00(79.8) | **22.40**(81.4) | 100.0(42.8) | 95.17(67.8) | **65.40**(74.2) | 100.0(41.4) | 46.00(71.8) | **38.80**(70.1) | |
| C-MNIST (FG) | 10 | 99.49(72.0) | 22.88(91.4) | **21.01**(91.7) | 100.0(26.2) | 15.02(94.0) | **14.74**(94.1) | 100.0(22.1) | 11.73(92.8) | 18.10(92.1) | 100.0(28.2) | **13.26**(92.8) | 17.45(92.1) | 10.06(97.69) |
| | 50 | 64.86(88.5) | 26.44(92.4) | **10.39**(95.3) | 100.0(59.6) | 10.57(96.1) | **8.10**(96.9) | 100.0(47.1) | 11.38(95.9) | **10.00**(96.1) | 100.0(42.5) | 17.55(94.2) | **17.35**(93.5) | |
| | 100 | 83.09(85.6) | 22.44(92.8) | **17.39**(95.1) | 100.0(71.1) | 8.12(97.0) | **7.62**(97.7) | 100.0(66.5) | 9.75(96.7) | **8.95**(97.1) | 100.0(34.3) | **14.67**(94.0) | 18.40(92.6) | |
| C-MNIST (BG) | 10 | 99.67(67.6) | 20.79(91.4) | **17.51**(91.3) | 100.0(20.3) | 15.12(94.7) | **12.99**(94.5) | 100.0(18.5) | 91.14(79.7) | **17.29**(89.6) | 100.0(19.3) | 18.13(91.6) | **13.13**(91.0) | 10.30(97.70) |
| | 50 | 58.91(89.0) | **19.77**(92.7) | 27.26(93.5) | 100.0(48.8) | 8.45(96.4) | **7.46**(96.8) | 100.0(47.6) | 90.39(81.2) | **17.18**(93.4) | 100.0(36.8) | **14.90**(93.9) | 17.10(92.3) | |
| | 100 | 94.55(87.3) | 56.09(92.4) | **30.09**(92.6) | 100.0(48.8) | 8.29(97.2) | **7.58**(97.0) | 100.0(66.1) | 97.45(80.0) | **13.29**(94.9) | 100.0(40.8) | **14.82**(92.4) | 41.79(80.5) | |
| UTKFace | 10 | 50.83(60.4) | 45.83(61.0) | **36.67**(62.8) | 54.17(66.8) | 40.17(67.5) | **38.33**(70.7) | 66.83(64.8) | 41.17(65.0) | **40.70**(66.1) | 61.40(55.9) | 49.20(60.5) | **45.20**(55.6) | 43.60(83.30) |
| | 50 | 51.83(71.0) | 35.67(70.9) | **29.00**(72.3) | 53.50(72.3) | 38.83(74.2) | **35.00**(74.8) | 51.67(69.8) | 43.33(71.7) | **40.30**(71.5) | 52.20(58.3) | 46.20(56.0) | **38.80**(57.9) | |
| | 100 | 47.83(70.8) | **26.50**(70.0) | 26.67(72.3) | 48.83(72.9) | **31.00**(76.0) | 32.33(75.9) | 46.67(72.6) | 41.00(73.8) | **36.83**(74.8) | 18.40(38.4) | 13.60(37.3) | **12.14**(37.4) | |
| BFFHQ | 10 | 48.80(61.3) | 43.80(63.8) | **32.87**(65.2) | 44.80(65.1) | 22.13(68.4) | **15.73**(69.5) | 37.20(63.5) | 22.33(63.8) | **19.56**(65.3) | 45.20(62.1) | 33.60(64.1) | **22.96**(66.3) | 64.00(72.66) |
| | 50 | 52.40(64.5) | 53.53(69.8) | **40.73**(71.6) | 60.27(62.9) | 24.13(72.6) | **15.13**(72.5) | 62.87(65.5) | 38.40(70.1) | **16.64**(70.7) | 35.76(59.6) | 33.52(63.6) | **23.68**(65.9) | |
| | 100 | 59.60(65.5) | 56.93(69.0) | **44.07**(71.1) | 63.47(65.8) | 22.53(74.4) | **7.87**(74.2) | 65.27(65.5) | 34.80(72.0) | **15.36**(74.7) | 7.12(51.1) | 5.60(50.2) | **5.28**(50.9) | |

*Table 2.* Fairness and accuracy as SKEW varies (top) on CIFAR10-S with fixed group GAP and IPC =50, and as group GAP varies (bottom) at fixed SKEW =0.8. Each cell reports EOD and, in parentheses, Acc for DC/DM/IDC distilled datasets (Vanilla, FairDD, COBRA), together with the FULL baseline.

| Method | Value | DC | | | DM | | | IDC | | | FULL |
|---|---|---|---|---|---|---|---|---|---|---|---|
| | | Vanilla | FairDD | COBRA | Vanilla | FairDD | COBRA | Vanilla | FairDD | COBRA | |
| SKEW | 0.60 | 32.24(44.6) | 13.64(45.7) | **10.16**(46.4) | 27.05(49.2) | 9.13(51.6) | **8.53**(54.0) | 34.50(43.0) | 10.22(46.7) | **8.08**(46.4) | 14.13(75.2) |
| | 0.65 | 45.10(43.7) | 14.48(45.8) | **11.28**(46.2) | 33.13(48.0) | 11.65(51.4) | **8.99**(52.8) | 44.87(42.1) | 12.48(47.1) | **8.40**(46.2) | 19.13(74.7) |
| | 0.70 | 57.29(41.4) | 15.00(45.4) | **11.33**(45.5) | 43.53(46.3) | 12.90(51.0) | **10.83**(52.9) | 55.53(39.8) | 12.23(46.1) | **7.43**(46.1) | 26.27(74.0) |
| | 0.75 | 65.81(39.2) | 19.55(44.8) | **15.51**(45.4) | 54.59(44.3) | 17.00(50.6) | **11.19**(52.5) | 65.23(37.3) | 13.73(46.8) | **7.82**(46.4) | 33.90(72.6) |
| | 0.80 | 70.43(37.6) | 26.05(44.5) | **14.70**(45.2) | 61.94(42.3) | 22.94(50.1) | **12.59**(52.6) | 70.63(36.1) | 16.63(47.2) | **7.65**(46.2) | 42.73(70.7) |
| | 0.85 | 76.38(35.9) | 32.96(43.4) | **17.68**(44.6) | 78.54(39.2) | 24.84(49.6) | **13.27**(51.6) | 73.13(34.6) | 14.65(46.2) | **7.65**(46.0) | 52.60(68.4) |
| GAP | 0 | 0.0(42.4) | 0.0(42.1) | 0.0(42.3) | 0.0(49.7) | 0.0(49.9) | 0.0(49.8) | 0.0(45.2) | 0.0(45.2) | 0.0(45.1) | 0.0(86.8) |
| | 1 | 30.93(37.1) | 10.15(41.8) | **7.98**(40.2) | 46.13(40.6) | 16.12(44.7) | **8.42**(44.5) | 41.87(37.2) | 7.98(42.4) | **4.63**(41.1) | 27.30(82.6) |
| | 2 | 50.65(34.5) | 27.12(40.0) | **11.32**(40.2) | 62.23(35.7) | 12.53(45.8) | **9.10**(44.9) | 53.77(36.3) | 10.10(41.9) | **9.90**(42.0) | 39.50(79.6) |
| | 3 | 56.75(30.8) | 27.75(36.7) | **11.73**(35.2) | 72.40(30.1) | 19.73(40.7) | **17.28**(41.2) | 56.53(34.2) | 17.80(39.6) | **12.98**(39.4) | 54.85(65.9) |
| | 4 | 61.40(27.4) | 38.22(33.3) | **16.38**(31.2) | 74.90(29.2) | 19.70(37.7) | **19.52**(37.3) | 57.43(31.1) | 24.80(35.9) | **18.95**(36.2) | 59.00(62.9) |

and progressively increase GAP from identical representations (GAP = 0) up to GAP = 4 (implementation details are in Appendix F). Table 2 shows that **both higher imbalance (SKEW) and higher GAP systematically worsen** EOD across all distillation baselines. As SKEW increases (with GAP fixed), Vanilla DD exhibits a near-monotonic rise in EOD for all backbones, while **fairness-aware methods mitigate this trend**. FairDD reduces EOD and CObra achieves the lowest EOD throughout. At mild imbalance (SKEW = 0.60), the COBRA and FairDD margins fall within overlapping standard deviations since as SKEW → 0.5 the uniform barycenter and the vanilla aggregate mixture coincide. The sweep should thus be interpreted in a directional manner, with the mechanism having its pri-

mary impact at higher levels of imbalance (for instance, at SKEW = 0.85 on DM, where COBRA attains 13.27 compared to 24.84 for FairDD).

With SKEW fixed at 0.8, increasing separation has an immediate impact. **When GAP = 0, EOD = 0 for all methods by construction**, but even mild separation (GAP = 1) sharply increases bias for Vanilla DD and larger GAP further intensifies it. FairDD and especially COBRA again provide strong mitigation, with **COBRA consistently delivering the best fairness outcomes**. Accuracy declines for all methods as GAP grows, aligning with the intuition that separation both **makes the task harder** and **amplifies group-dependent errors**. Overall, **imbalance induces a gradual** EOD **increase**, whereas **separation triggers a**

**more abrupt fairness degradation** once groups become distinguishable. Across both sweeps, **COBRA is the most robust**, achieving the lowest EOD while maintaining competitive accuracy. Standard deviations for all entries in Table 2, computed under the same 10-run protocol used for Table 1, are reported in Tables 13 and 12 of Appendix H.

### 5.3. Visualization Analysis

We investigate how different distillation objectives influence the learned representation with respect to both class labels and the sensitive attribute (group). Using C-MNIST (FG) with 10 classes and 10 demographic groups, we generate a t-SNE projection of features obtained from a classifier trained on the distilled dataset at IPC =50. Figure 3 presents the learned representations colored by class and by group, using a consistent color scheme across methods within each row. With DC, the class-colored embedding appears fragmented and partially entangled (Fig. 3a), while the group-colored embedding exhibits clearly separated, group-dependent regions (Fig. 3d), showing that the learned features still encode strong information about the sensitive groups. FairDD yields better class separation than DC (Fig. 3b); however, the group-colored representation remains visibly structured (Fig. 3e), with many clusters dominated by a single group color, indicating that sensitive information is attenuated but not fully eliminated. By contrast, COBRA produces compact, well-separated class clusters (Fig. 3c) and exhibits markedly stronger mixing of group colors within these clusters (Fig. 3f). This behavior aligns with COBRA's objective of distilling toward a subgroup-agnostic barycentric target within each class, which suppresses subgroup-specific components while maintaining class-discriminative structure.

### 5.4. Generalization Across Architectures

A key requirement for dataset distillation is *cross-architecture generalization*. A distilled dataset should remain effective when training models whose architectures differ from the one used during distillation. Strong transfer across architectures suggests that the distilled images encode task-relevant, model-agnostic information rather than leveraging architecture-specific inductive biases or optimization artifacts. An analogous condition applies to *fairness*. Any fairness gains introduced by distillation should also generalize across architectures; otherwise, they risk reflecting architecture-specific artifacts instead of providing stable bias mitigation. To evaluate this, we distill a single labeled set $S$ with a fixed IPC budget using DM, employing ConvNet as the distillation model, and then train a variety of networks from scratch on $S$. Our evaluation suite spans architectures with different capacities and designs, including AlexNet (Krizhevsky et al., 2012), VGG11 (Simonyan & Zisserman, 2015), and ResNet18 (He et al., 2016).

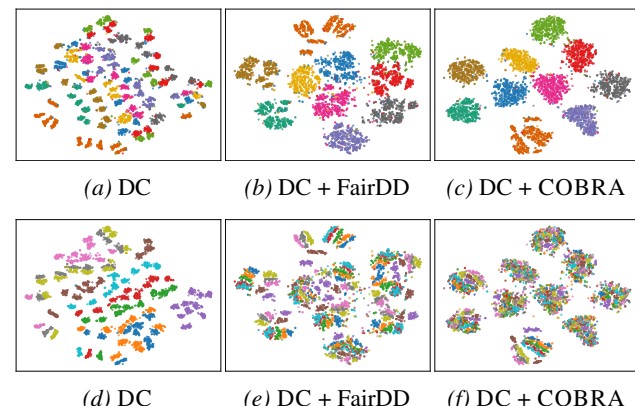

*(a)* DC     *(b)* DC + FairDD     *(c)* DC + COBRA

*(d)* DC     *(e)* DC + FairDD     *(f)* DC + COBRA

*Figure 3.* t-SNE visualization of last-layer representations on C-MNIST (FG), IPC =50. The top row colors points by class, while the bottom row colors points by sensitive attribute. Color mappings are uniform across methods within each row.

As shown in Table 3, the COBRA distilled set *consistently* outperforms the vanilla DM baseline across all evaluation architectures. For every dataset, COBRA attains the lowest EOD while simultaneously achieving higher Acc, and this ranking is preserved when evaluated on different models. The small cross-model variation further indicates that these gains are **stable** rather than dependent on a specific architectural inductive bias, demonstrating robust transfer of both accuracy and fairness improvements across networks.

### 5.5. Ablation on Barycenter Discrepancy

To isolate the specific role of the barycentric discrepancy, we replaced the particular barycenter discrepancy employed in Section 5.1 with several alternative distance measures. Concretely, we resolved (8) with $d$ instantiated as $\ell_1$, $\ell_2$, $\ell_\infty$, cosine, and Huber (Huber, 1992). Details for each choice of $d$ are provided in Appendix D. In all cases, we optimized $m$ using the L-BFGS (Liu & Nocedal, 1989), initializing $m$ at the mean of the subgroup statistics and then minimizing $\sum_a d(\Phi_a, m)$. Table 4 shows that the choice of discrepancy $d$ can shift fairness, but the effect is highly non-uniform across datasets. In some cases, alternative discrepancies surpass our default (e.g., CIFAR10-S), whereas in others COBRA is superior (C-MNIST (FG)). Moreover, no single discrepancy performs best across all benchmarks. While $\ell_\infty$ and cosine improve results on C-FMNIST, they significantly harm performance on BFFHQ. Although $\ell_2$ benefits C-MNIST (BG), it underperforms on CIFAR10-S. Overall, substituting our default discrepancy with a "more sophisticated" choice fails to yield robust fairness gains; improvements on one dataset are often offset by degradations on another. This pattern suggests that fairness gains mainly arise from barycentric alignment, while the geometry defined by $d$ acts as a dataset-specific hyperparameter.

*Table 4.* Ablation on discrepancy $d$ (DM, IPC = 10). Cells show EOD after recomputing the barycenter using the specified $d$.

| Dataset | $\ell_1$ | $\ell_2$ | $\ell_\infty$ | Cosine | Huber | COBRA |
|---|---|---|---|---|---|---|
| CIFAR10-S | 16.35 | 20.99 | 18.60 | 16.96 | 15.36 | 20.18 |
| C-FMNIST (FG) | 36.00 | 41.33 | 27.17 | 27.00 | 32.17 | 25.83 |
| C-FMNIST (BG) | 47.17 | 49.83 | 38.00 | 36.33 | 38.83 | 33.80 |
| C-MNIST (FG) | 44.57 | 26.63 | 18.52 | 21.16 | 18.28 | 14.74 |
| C-MNIST (BG) | – | 12.78 | 17.00 | 15.03 | 13.57 | 12.99 |
| UTKFACE | 37.83 | 43.00 | 34.83 | 37.83 | 39.67 | 38.33 |
| BFFHQ | 19.47 | 19.73 | 30.00 | 32.53 | 25.60 | 15.73 |

*Table 5.* Worst-case residual geometry for validating Theorem 4.1. Each entry reports $\mathrm{MaxRes_{COBRA}}$ averaged over classes, with the worst class (maximum) in parentheses. Bold marks the lowest mean within each block (same objective and IPC).

| Objective | Dataset | Vanilla | FairDD | COBRA |
|---|---|---|---|---|
| DC | BFFHQ | 117.891 (198.667) | 41.245 (57.090) | **15.454 (21.730)** |
| | BFFHQ | 254.052 (277.810) | 105.988 (148.130) | **22.805 (33.700)** |
| | CIFAR10-S | 59.267 (103.680) | 46.417 (80.887) | **40.012 (68.493)** |
| | CIFAR10-S | 403.143 (776.124) | 91.911 (139.054) | **79.756 (134.649)** |
| DM | BFFHQ | 270.327 (357.298) | 50.648 (57.988) | **45.250 (54.682)** |
| | BFFHQ | 557.932 (750.967) | 53.672 (69.452) | **45.235 (54.819)** |
| | CIFAR10-S | 171.403 (365.839) | 49.884 (87.755) | **42.236 (72.801)** |
| | CIFAR10-S | 580.187 (1108.934) | 58.918 (101.583) | **42.258 (61.716)** |

## 5.6. Ablation on Worst-Case Residual Bound

Here, we perform an ablation study to further substantiate the claims in Theorem 4.1. For each method and for each IPC, we train a model on the distilled dataset until convergence and then compute the subgroup statistics $\Phi_{a|y}$ on the real data for every class $y$. We define $\phi_{a|y}$ as class-conditional subgroup gradients for DC and as subgroup mean embeddings for DM. Using these, we form the vanilla DD and FairDD targets $m_y^{\mathrm{van}}$, as well as the CO-BRA barycenter target $m_y^*$, and compare their worst-case residuals $\max_a \|\phi_{a|y} - m\|_2^2$. We present both the mean and the worst-class residuals in Table 5. Across selected datasets and distillation objectives, the COBRA target consistently achieves smaller worst-case subgroup residuals than the Vanilla DD mixture target, often by a wide margin, and it also improves upon the FairDD variant. This effect is most pronounced for Vanilla DD, where the class-averaged $\mathrm{MaxRes_{van}}$ is substantially larger than for either FairDD or COBRA, indicating that naive targets can be severely misaligned with the underlying subgroup geometry. These findings are consistent with the observation that, in all of these settings, COBRA attains the lowest EOD, as reported in Table 1. Additional details are provided in Appendix G.2.

## 5.7. Image Visualization & Additional Results

Figure 4 illustrates how COBRA qualitatively modifies the distilled set. With standard DM, backgrounds are nearly uniform within each class, indicating that the synthetic prototypes still encode the spurious group signal. By contrast,

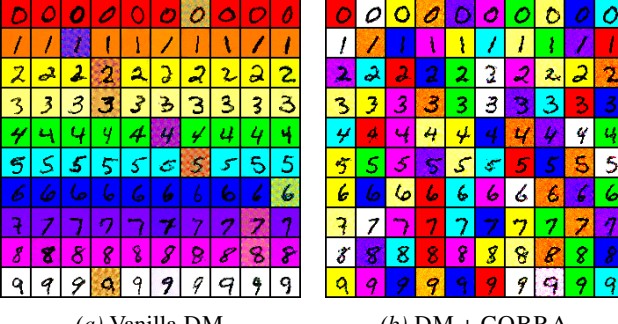

*(a)* Vanilla DM       *(b)* DM + COBRA

*Figure 4.* Distilled C-MNIST (BG) at IPC = 10 using Vanilla DM and COBRA.

COBRA promotes greater within-class background diversity while maintaining well-defined digit shapes, consistent with the gains in subgroup fairness reported in our quantitative analysis. Further examples are given in Appendix M.

## 5.8. Ablation on MTT

We next evaluate whether COBRA can also be applied to trajectory-based dataset distillation. This setup is especially informative because MTT (Cazenavette et al., 2022) does not align class-level statistics such as gradients or embeddings; instead, it aligns training dynamics in parameter space. Specifically, MTT first computes *expert trajectories* by training networks on the full dataset and recording parameter snapshots over epochs. During distillation, it samples a start epoch $t$ from one expert trajectory, initializes a student

*Table 3.* Cross-architecture generalization using DM at IPC = 50. Each cell is reported as EOD *(Acc)*. The best-performing results for each dataset and architecture are shown in **bold**.

| Model | CIFAR10-S | | C-FMNIST (FG) | | C-MNIST (BG) | | UTKFACE | | BFFHQ | |
|---|---|---|---|---|---|---|---|---|---|---|
| | Vanilla | COBRA | Vanilla | COBRA | Vanilla | COBRA | Vanilla | COBRA | Vanilla | COBRA |
| ConvNet | 73.88 (45.5) | **15.82 (57.6)** | 100.0 (50.6) | **19.60 (82.9)** | 100.0 (48.8) | **7.54 (96.9)** | 53.50 (71.83) | **33.50 (74.4)** | 59.40 (63.27) | **15.60 (72.6)** |
| AlexNet | 73.34 (35.6) | **15.22 (46.2)** | 100.0 (37.9) | **25.40 (80.8)** | 60.0 (13.5) | **9.74 (96.5)** | 49.25 (66.31) | **36.50 (72.9)** | 54.0 (64.62) | **11.30 (74.2)** |
| VGG11 | 61.88 (43.3) | **10.24 (52.4)** | 100.0 (58.7) | **17.80 (83.0)** | 100.0 (66.0) | **7.75 (97.0)** | 55.75 (67.69) | **37.50 (70.0)** | 51.80 (65.15) | **8.70 (69.8)** |
| ResNet18 | 74.26 (38.4) | **15.72 (50.4)** | 100.0 (42.3) | **21.80 (81.6)** | 100.0 (26.2) | **7.34 (97.4)** | 56.25 (66.60) | **42.50 (67.7)** | 55.40 (63.12) | **10.20 (66.9)** |
| Mean | 70.84 (40.7) | 14.25 (51.7) | 100.0 (47.4) | 21.15 (82.1) | 90.00 (38.6) | 8.09 (97.0) | 53.69 (68.1) | 37.50 (71.3) | 55.15 (64.0) | 11.45 (70.9) |
| STD | 5.18 (3.9) | 2.33 (4.1) | 0.0 (8.0) | 2.83 (0.9) | 17.32 (20.2) | 0.96 (0.32) | 2.76 (2.2) | 3.24 (2.6) | 2.77 (0.9) | 2.57 (2.8) |

network from that checkpoint, performs $K$ gradient steps on the synthetic data, and then updates the synthetic images so that the resulting student parameters match the expert's parameters at epoch $t + K$.

To integrate COBRA into MTT, we preserve the original optimization loop and modify only the construction of real-data targets. Instead of a single replay buffer over all data, we maintain one buffer per demographic group. At each distillation step, we sample one trajectory from each group, all initialized at the same epoch $t$, and form cross-group barycentric start and end points by averaging the corresponding checkpoints:

$$\bar{\theta}_t = \frac{1}{G} \sum_{g=1}^{G} \theta_{g,t}, \qquad \bar{\theta}_{t+K} = \frac{1}{G} \sum_{g=1}^{G} \theta_{g,t+K},$$

where $G$ denotes the total number of groups. We initialize the student at $\bar{\theta}_t$ and then optimize the synthetic dataset so that, after $K$ synthetic updates, the student parameters match $\bar{\theta}_{t+K}$. Consequently, COBRA-MTT is a lightweight adaptation of standard MTT that replaces a single aggregate trajectory target with a group-balanced, barycentric trajectory target.

Table 6 demonstrates that COBRA remains effective under this fundamentally different distillation signal. On BFFHQ, COBRA-MTT lowers EOD from 32.96% to 9.12%, indicating a substantial fairness improvement. This improvement comes with a drop in accuracy from 61.64% to 51.86%. This indicates that vanilla MTT can retain group-specific shortcuts that improve overall accuracy while causing large subgroup disparities, whereas barycentric trajectory matching mitigates these shortcuts. On CIFAR10-S, COBRA-MTT improves both metrics, raising accuracy from 26.36% to 36.32% and reducing EOD from 39.40% to 30.38%. Collectively, these findings indicate that COBRA is not limited to representation-matching objectives and can be incorporated into trajectory matching with only minor modifications to the target construction.

*Table 6.* COBRA-MTT with IPC $= 10$. Values are reported over five runs. The superior value for each method is bold.

| Method | Dataset | Accuracy | EOD |
|---|---|---|---|
| Vanilla-MTT | BFFHQ | **61.64** $\pm$ 1.09 | 32.96 $\pm$ 6.55 |
| COBRA-MTT | BFFHQ | 51.86 $\pm$ 1.71 | **9.12** $\pm$ 1.95 |
| Vanilla-MTT | CIFAR10-S | 26.36 $\pm$ 0.21 | 39.40 $\pm$ 1.77 |
| COBRA-MTT | CIFAR10-S | **36.32** $\pm$ 0.58 | **30.38** $\pm$ 2.58 |

**Additional analyses.** Supplementary experiments covering low-IPC robustness for IPC $\in \{1, 3, 5\}$, ablations comparing our COBRA with standard fairness interventions, performance under partially observed group labels, and an assessment of computational overhead are presented in Appendices G.3, L, K, and I.

## 6. Conclusion

Dataset distillation is starting to be used in settings where subgroup disparities matter; yet, most distillation objectives are still optimized for average accuracy. In this work, we show that subgroup unfairness in distilled data is not explained by group imbalance alone or by subgroup representation differences alone. Instead, unfairness is driven by their interaction, where an aggregate distillation target can be dominated by the majority group while also pulling the synthetic set toward group-specific representation patterns. We address this with COBRA, a fairness-oriented objective that forms a shared target across subgroups before computing the distillation loss, rather than only reweighting subgroup losses after the fact. COBRA is compatible with existing distillation pipelines and can be applied without changing the downstream training procedure. Across multiple biased benchmarks and distillation backbones, COBRA consistently reduces subgroup disparity while maintaining competitive accuracy. These gains persist under low IPC budgets and transfer across architectures, which supports the view that the distilled samples capture more stable, group-robust information. Future directions include extending barycentric targets to settings with missing or noisy group annotations, multiple protected attributes, and distribution shifts between distillation and deployment.

## Acknowledgements

The authors acknowledge the Vector Institute for providing computational resources. This research was supported by the Natural Sciences and Engineering Research Council of Canada (NSERC) through the Discovery Grants program including grant RGPIN-2025-04585 and by the John Thompson Chair Fellowship.

## Impact Statement

This work addresses fairness in dataset distillation, which compresses large datasets into small synthetic ones. While distillation can reduce storage and training costs, it may also amplify subgroup disparities when the distilled data underrepresents demographic-specific predictive patterns. COBRA aims to mitigate this risk by aligning distilled representations with a subgroup-agnostic barycentric target, improving fairness while preserving predictive performance.

The potential benefit of this work is to make dataset distillation more reliable for fairness-sensitive applications such as healthcare, finance, and other high-stakes settings. However, the method relies on group annotations during distillation, which may be unavailable or noisy. COBRA should therefore be used within a broader fairness evaluation process that includes domain-specific auditing, appropriate fairness metrics, and monitoring in real deployment.

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

# A. Appendix

# B. Related Work

**Coresets**   Coresets are weighted subsets of the data that can be used to train models with nearly the same performance as training on the full dataset. Many selection methods exist, but most are task-specific, meaning they are tailored to a particular learning model family. Examples include coresets for $k$-means and clustering (Jubran et al., 2020; Cohen-Addad et al., 2022; Har-Peled & Mazumdar, 2004), regression (Meyer et al., 2022; Maalouf et al., 2019), mixture models (Lucic et al., 2018), and Bayesian inference (Campbell & Broderick, 2018). In deep learning, recent methods build coresets by aligning the gradients of the full dataset with weighted subset (Mirzasoleiman et al., 2020; Killamsetty et al., 2021; Tukan et al., 2023). However, selecting a small subset that preserves performance can still be difficult in high-dimensional settings or large datasets.

**Fairness-aware coresets**   Fairness constraints have been incorporated into coreset design, most prominently for fair clustering (Schmidt et al., 2019; Huang et al., 2019; Bandyapadhyay et al., 2024; Liang et al., 2025). Streaming fair coresets produce compact weighted summaries that can be merged across data batches, supporting scalable fair $k$-means in streaming scenarios (Schmidt et al., 2019). In the presence of multiple, possibly overlapping sensitive attributes, one can construct coresets by partitioning the data points based on their combinations of group memberships and then selecting weighted representative points, ensuring that the fair $k$-means objectives are well approximated (Huang et al., 2019). Sampling-based constructions yield fair coresets in arbitrary metric spaces and avoid exponential dimension dependence in Euclidean settings, leading to faster approximation and streaming methods (Bandyapadhyay et al., 2024). Beyond clustering, fairness-aware coresets for regression and individually fair clustering modify both the coreset definition and sampling probabilities to preserve loss and fairness constraints (Chhaya et al., 2022). Applied work includes interactive, human-in-the-loop selection of fairness-aware coresets with explanations for tabular data (Hadar et al., 2025), and optimal-transport-based representatives via fair Wasserstein coresets (Xiong et al., 2024). This line of work is *distinct from fairness in dataset distillation*, which explicitly learns *synthetic* training examples that each summarize many original samples via matched training dynamics, whereas coresets instead select *real* datapoints (with weights), so every chosen point corresponds to one original example rather than a learned prototype.

**Fairness-aware Data Generation**   Another line of research incorporates fairness constraints *directly into the process of generating synthetic data*. FairGAN (Xu et al., 2018) alters GAN training by adding an explicit fairness objective so that the synthesized data stay realistic while reducing the dependence between protected attributes and target outcomes. Fairness GAN (Sattigeri et al., 2019) extends an auxiliary-classifier GAN with fairness regularizer for criteria such as demographic parity and equality of opportunity, yet still produces realistic, high-dimensional samples. Instead of producing a completely new dataset, data-domain fair representation learning applies a structured modification to each individual example, (Quadrianto et al., 2019) formulates fairness as data-to-data translation, generating representations that lie in the original input space and thus allow qualitative examination of what semantic alterations are needed to satisfy a chosen fairness notion. This is again distinct from dataset distillation. Fair data generation aims to approximate the original data distribution while improving fairness at the *distribution* level and can yield arbitrarily many samples, whereas distillation learns a *small* synthetic training set tuned to mimic the training dynamics, with each synthetic instance effectively summarizing numerous real examples.

# C. Proof of Theorems

## C.1. Setting and notation

We fix a class label set $\mathcal{Y}$ and a finite sensitive-attribute (subgroup) set $\mathcal{A}$ with $|\mathcal{A}| < \infty$. For a fixed class $y \in \mathcal{Y}$, let $T_{a|y}$ denote the class-conditional subgroup corresponding to subgroup $a \in \mathcal{A}$ within class $y$. Let $\pi_{a|y} \geq 0$ be the class-conditional subgroup weights, satisfying $\sum_{a \in \mathcal{A}} \pi_{a|y} = 1$ (e.g., $\pi_{a|y} = \Pr(A = a \mid Y = y)$ in the population, or the empirical fraction within class $y$). Let $Q \in \mathbb{R}^{d \times d}$ be symmetric positive definite (SPD), denoted $Q \succ 0$. Define the $Q$-inner product and induced norm by

$$\langle u, v \rangle_Q := u^\top Q v, \qquad \|u\|_Q := \sqrt{u^\top Q u},$$

and the corresponding squared distance

$$d(u, v) := \|u - v\|_Q^2 = (u - v)^\top Q(u - v).$$

**Vanilla and COBRA class targets.** For each class $y \in \mathcal{Y}$, define:

$$m_y^{\text{van}} := \sum_{a \in \mathcal{A}} \pi_{a|y}\, \Phi_{T_{a|y}}, \tag{11}$$

$$m_y^{\star} \in \arg \min_{m \in \mathbb{R}^d} \frac{1}{|\mathcal{A}|} \sum_{a \in \mathcal{A}} d\big(\Phi_{T_{a|y}}, m\big). \tag{12}$$

The point $m_y^{\star}$ is the (uniform-weight) barycenter used by COBRA under the distance $d$.

**Residuals and the center shift.** Given any center $m \in \mathbb{R}^d$, define the class-conditional subgroup residual for subgroup $a$ as $\Delta_{a|y}(m) := \Phi_{T_{a|y}} - m$. In particular, for COBRA and Vanilla:

$$\Delta_{a|y}^{\text{C}} := \Delta_{a|y}(m_y^{\star}), \qquad \Delta_{a|y}^{\text{V}} := \Delta_{a|y}(m_y^{\text{van}}).$$

We also define the *center shift*

$$s_y := m_y^{\text{van}} - m_y^{\star}.$$

Note the identity:

$$\Delta_{a|y}^{\text{V}} = \Phi_{T_{a|y}} - m_y^{\text{van}} = \Phi_{T_{a|y}} - m_y^{\star} - (m_y^{\text{van}} - m_y^{\star}) = \Delta_{a|y}^{\text{C}} - s_y. \tag{13}$$

## C.2. Closed-form barycenter for squared $Q$-norm

**Proposition C.1.** *Assume $d(u,v) = \|u - v\|_Q^2$ for some $Q \succ 0$. Then for any fixed class $y \in \mathcal{Y}$, the inner problem* (12) *has a unique minimizer given by the uniform mean of subgroup statistics:*

$$m_y^{\star} = \frac{1}{|\mathcal{A}|} \sum_{a \in \mathcal{A}} \Phi_{T_{a|y}}.$$

*Proof.* Fix $y \in \mathcal{Y}$ and abbreviate $\phi_a := \Phi_{T_{a|y}} \in \mathbb{R}^d$ for $a \in \mathcal{A}$. Consider the objective

$$f(m) := \frac{1}{|\mathcal{A}|} \sum_{a \in \mathcal{A}} \|\phi_a - m\|_Q^2 = \frac{1}{|\mathcal{A}|} \sum_{a \in \mathcal{A}} (\phi_a - m)^\top Q (\phi_a - m).$$

Expanding and using symmetry of $Q$,

$$f(m) = \frac{1}{|\mathcal{A}|} \sum_{a \in \mathcal{A}} \left( \phi_a^\top Q \phi_a - 2 m^\top Q \phi_a + m^\top Q m \right)$$

$$= \underbrace{\frac{1}{|\mathcal{A}|} \sum_{a \in \mathcal{A}} \phi_a^\top Q \phi_a}_{\text{constant in } m} - \frac{2}{|\mathcal{A}|} m^\top Q \sum_{a \in \mathcal{A}} \phi_a + m^\top Q m.$$

Taking derivatives with respect to $m$ yields

$$\nabla f(m) = 2 Q m - \frac{2}{|\mathcal{A}|} Q \sum_{a \in \mathcal{A}} \phi_a = \frac{2}{|\mathcal{A}|} Q \left( |\mathcal{A}| m - \sum_{a \in \mathcal{A}} \phi_a \right).$$

Setting $\nabla f(m) = 0$ and using that $Q$ is invertible (since $Q \succ 0$) gives

$$|\mathcal{A}| m = \sum_{a \in \mathcal{A}} \phi_a \implies m = \frac{1}{|\mathcal{A}|} \sum_{a \in \mathcal{A}} \phi_a.$$

To show uniqueness, observe that the Hessian is

$$\nabla^2 f(m) = 2Q \succ 0,$$

Hence, $f$ is strictly convex, which implies that the stationary point is the unique global minimizer. Replacing $\phi_a$ with $\Phi_{T_{a|y}}$ then establishes the claim. $\qquad\square$

## C.3. Worst-case subgroup residual comparison

**Theorem C.2.** *Fix $y \in \mathcal{Y}$ and assume $d(u,v) = \|u - v\|_Q^2$ with $Q \succ 0$. Let*

$$m_y^\star = \frac{1}{|\mathcal{A}|} \sum_{a \in \mathcal{A}} \Phi_{T_{a|y}} \qquad and \qquad m_y^{\mathrm{van}} = \sum_{a \in \mathcal{A}} \pi_{a|y} \Phi_{T_{a|y}},$$

*be the COBRA and Vanilla targets, respectively. Define residuals*

$$\Delta_{a|y}^{\mathrm{C}} := \Phi_{T_{a|y}} - m_y^\star, \qquad \Delta_{a|y}^{\mathrm{V}} := \Phi_{T_{a|y}} - m_y^{\mathrm{van}},$$

*and the shift $s_y := m_y^{\mathrm{van}} - m_y^\star$. Assume there exists $a^\dagger \in \arg\max_{a \in \mathcal{A}} \|\Delta_{a|y}^{\mathrm{C}}\|_Q$ such that the following condition holds:*

$$\langle \Delta_{a^\dagger|y}^{\mathrm{C}}, s_y \rangle_Q \leq 0.$$

*Then COBRA does not increase the worst-case subgroup residual:*

$$\max_{a \in \mathcal{A}} \|\Delta_{a|y}^{\mathrm{C}}\|_Q \leq \max_{a \in \mathcal{A}} \|\Delta_{a|y}^{\mathrm{V}}\|_Q.$$

*Proof.* Fix $y \in \mathcal{Y}$ and abbreviate $\Delta_a^{\mathrm{C}} := \Delta_{a|y}^{\mathrm{C}}$, $\Delta_a^{\mathrm{V}} := \Delta_{a|y}^{\mathrm{V}}$, and $s := s_y$ to simplify notation. By the shift identity (13),

$$\Delta_a^{\mathrm{V}} = \Delta_a^{\mathrm{C}} - s \qquad \text{for all } a \in \mathcal{A}. \tag{*}$$

Let $a^\dagger \in \arg\max_{a \in \mathcal{A}} \|\Delta_a^{\mathrm{C}}\|_Q$ be an index that attains the maximum COBRA residual and satisfies the condition $\langle \Delta_{a^\dagger}^{\mathrm{C}}, s \rangle_Q \leq 0$.

**Step 1: Compare the residual norms at $a^\dagger$.** Using (*) and expanding the squared $Q$-norm,

$$\begin{aligned}
\|\Delta_{a^\dagger}^{\mathrm{V}}\|_Q^2 &= \|\Delta_{a^\dagger}^{\mathrm{C}} - s\|_Q^2 \\
&= (\Delta_{a^\dagger}^{\mathrm{C}} - s)^\top Q (\Delta_{a^\dagger}^{\mathrm{C}} - s) \\
&= (\Delta_{a^\dagger}^{\mathrm{C}})^\top Q \Delta_{a^\dagger}^{\mathrm{C}} + s^\top Q s - 2(\Delta_{a^\dagger}^{\mathrm{C}})^\top Q s \\
&= \|\Delta_{a^\dagger}^{\mathrm{C}}\|_Q^2 + \|s\|_Q^2 - 2\langle \Delta_{a^\dagger}^{\mathrm{C}}, s \rangle_Q.
\end{aligned} \tag{**}$$

By assumption, $\langle \Delta_{a^\dagger}^{\mathrm{C}}, s \rangle_Q \leq 0$, hence $-2\langle \Delta_{a^\dagger}^{\mathrm{C}}, s \rangle_Q \geq 0$. Together with $\|s\|_Q^2 \geq 0$, equation (**) implies

$$\|\Delta_{a^\dagger}^{\mathrm{V}}\|_Q^2 \geq \|\Delta_{a^\dagger}^{\mathrm{C}}\|_Q^2 \qquad \Longrightarrow \qquad \|\Delta_{a^\dagger}^{\mathrm{V}}\|_Q \geq \|\Delta_{a^\dagger}^{\mathrm{C}}\|_Q. \tag{***}$$

**Step 2: Lift the pointwise comparison to a max comparison.** Since $a^\dagger$ attains the maximum COBRA residual,

$$\max_{a \in \mathcal{A}} \|\Delta_a^{\mathrm{C}}\|_Q = \|\Delta_{a^\dagger}^{\mathrm{C}}\|_Q.$$

Also, by definition of the maximum,

$$\max_{a \in \mathcal{A}} \|\Delta_a^{\mathrm{V}}\|_Q \geq \|\Delta_{a^\dagger}^{\mathrm{V}}\|_Q.$$

Combining these with (***) yields

$$\max_{a \in \mathcal{A}} \|\Delta_a^{\mathrm{V}}\|_Q \geq \|\Delta_{a^\dagger}^{\mathrm{V}}\|_Q \geq \|\Delta_{a^\dagger}^{\mathrm{C}}\|_Q = \max_{a \in \mathcal{A}} \|\Delta_a^{\mathrm{C}}\|_Q,$$

which is exactly (C.2).

$\square$

## D. Alternative Discrepancies for Barycenter

In contrast to squared Mahalanobis discrepancies, the following common choices generally *do not* admit the uniform arithmetic mean as a unique minimizer of the barycenter objective, because the objective is either non-smooth (no unique first-order stationary), or smooth but non-quadratic.

- $\ell_1$ **discrepancy.**

$$D_{\ell_1}(\{\Phi_a\}, m) \;=\; \frac{1}{G}\sum_{a=1}^{G}\|\Phi_a - m\|_1 \;=\; \frac{1}{G}\sum_{a=1}^{G}\sum_{i=1}^{d}\big|(\Phi_a - m)_i\big|.$$

  This objective is convex but non-smooth; minimizers are coordinate-wise medians (generally not the arithmetic mean).

- $\ell_2$ **discrepancy.**

$$D_{\ell_2}(\{\Phi_a\}, m) \;=\; \frac{1}{G}\sum_{a=1}^{G}\|\Phi_a - m\|_2.$$

  This objective is convex but non-quadratic; the optimality condition involves normalized residuals $\sum_a \frac{m-\Phi_a}{\|m-\Phi_a\|_2} = 0$, yielding the geometric median rather than the mean.

- $\ell_\infty$ **discrepancy.**

$$D_{\ell_\infty}(\{\Phi_a\}, m) \;=\; \frac{1}{G}\sum_{a=1}^{G}\|\Phi_a - m\|_\infty \;=\; \frac{1}{G}\sum_{a=1}^{G}\max_{1\le i\le d}\big|(\Phi_a - m)_i\big|.$$

  This objective is convex but non-smooth due to the max; minimizers are typically Chebyshev/minimax-type centers and need not coincide with the mean.

- **Cosine discrepancy.** Let $\cos(\Phi_a, m) = \frac{\langle \Phi_a, m\rangle}{\|\Phi_a\|_2 \|m\|_2}$.

$$D_{\cos}(\{\Phi_a\}, m) \;=\; \frac{1}{G}\sum_{a=1}^{G}\big(1 - \cos(\Phi_a, m)\big) \;=\; \frac{1}{G}\sum_{a=1}^{G}\left(1 - \frac{\langle \Phi_a, m\rangle}{\|\Phi_a\|_2 \|m\|_2}\right).$$

  This objective is scale-invariant in $m$ and non-convex; the optimizer prioritizes directional alignment with the $\Phi_a$'s rather than Euclidean averaging.

- **Huber discrepancy.** Define the (scalar) Huber penalty

$$h_\delta(t) = \begin{cases} \frac{1}{2}t^2, & |t| \le \delta, \\ \delta|t| - \frac{1}{2}\delta^2, & |t| > \delta, \end{cases}$$

applied elementwise to coordinates. Then

$$D_{\text{Huber}}(\{\Phi_a\}, m) \;=\; \frac{1}{G}\sum_{a=1}^{G}\sum_{i=1}^{d} h_\delta\big((\Phi_a - m)_i\big).$$

  While convex and piecewise-smooth, the first-order condition is a robust M-estimator with clipped residuals; the barycenter generally differs from the mean except in the purely quadratic regime $|(\Phi_a - m)_i| \le \delta$.

## E. Datasets

We evaluate methods under controlled spurious correlations between the classification label $y$ and a *sensitive attribute* $a$ (i.e., group membership). For each dataset, we construct a biased training split by assigning each class a designated majority group and sampling a fraction of that class from this group. We quantify the correlation strength using the **SKEW**, defined as the fraction of training samples belonging to the majority sensitive group (reported either as a single value when uniform across classes, or per-class when it varies). All test splits are **group-balanced**, meaning that for each class, the samples are uniformly distributed across the different sensitive groups. Further information is provided in Table 7. We now discuss the characteristics of each dataset in greater detail.

- **Colored MNIST.** The foreground version "C-MNIST (FG)" correlates $y$ (digit identity) with the *foreground color* $a$ by rendering digits in one dominant color per class (ten colors total), while the remaining samples are distributed across the other colors. The background version "C-MNIST (BG)" instead correlates $y$ with the *background color* $a$, leaving the digit foreground unchanged. In both cases, the test set is generated by uniformly applying all colors per class to remove correlation.

- **Colored Fashion-MNIST.** The foreground version "C-FMNIST (FG)" correlates $y$ (object category) with the *object color* $a$ using the same construction as C-MNIST (FG). The background version "C-FMNIST (BG)" correlates $y$ with the *background color* $a$ analogously to C-MNIST (BG). Test splits are group-balanced across colors.

- **CIFAR10-S.** This dataset introduces a binary sensitive attribute $a \in \{\text{color}, \text{grayscale}\}$ by converting a portion of the images to grayscale in a class-dependent manner, inducing a correlation between $y$ and $a$ during training. For evaluation, we duplicate the original CIFAR-10 test set, apply grayscale to the copy, and concatenate both halves, producing a group-balanced test split with doubled size.

- **UTKFACE.** This is an in-the-wild face dataset with over 20,000 images and annotations for age, gender, and race, covering a wide age span and substantial variation in pose, illumination, and occlusion. We partition age into three classes and use race as the sensitive attribute, utilizing four major race groups from the provided race annotations.

- **BFFHQ.** This is a gender-biased derivative of the FFHQ face dataset, constructed such that age serves as the prediction target while gender acts as a correlated bias. In particular, BFFHQ defines two age classes, "Young" and "Old", with typical age ranges 10–29 and 40–59, and a strong training correlation where Young is predominantly female and Old is predominantly male. In our setup, the task is age classification with two classes, and the sensitive attribute is gender. Unlike the synthetic color datasets, these correlations arise from the data, and the effective SKEW can vary by class.

**Condensation ratio.** When presenting condensation results, we adopt IPC $\in \{10, 50, 100\}$ as the **condensation ratio**, defined as $\frac{\text{IPC} \cdot |\mathcal{Y}|}{|\mathcal{D}_{\text{train}}|}$, where $|\mathcal{Y}|$ denotes the total number of classes.

*Table 7.* Overview of the datasets and group-level statistics used in our experiments. SKEW indicates the proportion of training examples belonging to the majority sensitive group (reported per class when it is not constant). All test splits are balanced across groups. The condensation ratio is reported for IPC values $\in \{10, 50, 100\}$.

| Dataset | Task label $y$ | Sensitive attribute $a$ | $|\mathcal{Y}|$ | $|\mathcal{A}|$ | Train | Test | SKEW | Test split | Condensation ratio (%) | | |
|---|---|---|---|---|---|---|---|---|---|---|---|
| | | | | | | | | | IPC=10 | IPC=50 | IPC=100 |
| C-MNIST (FG) | digit identity | foreground color | 10 | 10 | 60000 | 10000 | 0.90 | balanced | 0.17 | 0.83 | 1.67 |
| C-MNIST (BG) | digit identity | background color | 10 | 10 | 60000 | 10000 | 0.90 | balanced | 0.17 | 0.83 | 1.67 |
| C-FMNIST (FG) | object category | object color | 10 | 10 | 60000 | 10000 | 0.90 | balanced | 0.17 | 0.83 | 1.67 |
| C-FMNIST (BG) | object category | background color | 10 | 10 | 60000 | 10000 | 0.90 | balanced | 0.17 | 0.83 | 1.67 |
| CIFAR10-S | object category | grayscale indicator | 10 | 2 | 50000 | 20000 | 0.90 | balanced | 0.20 | 1.00 | 2.00 |
| UTKFACE | age | race | 3 | 4 | 20813 | 1200 | 0.53 / 0.35 / 0.63 | balanced | 0.14 | 0.72 | 1.44 |
| BFFHQ | age | gender | 2 | 2 | 19200 | 1000 | 0.995 / 0.995 | balanced | 0.10 | 0.52 | 1.04 |

## F. Details of Group-Imbalance and Separation-Effect Experiment

In this section, we provide additional details on the experiment described in Section 5.2. For the group-imbalance sweep (varying SKEW), we use the same variant of CIFAR10-S introduced in Appendix E, but increase the ratio of group imbalance. For the increasing group-separation setting (fixing SKEW and increasing GAP), we introduce a *hierarchical corruption severity* parameter $\alpha$ that specifies a cumulative sequence of image perturbations. Given a clean image $x$, we construct a corrupted image $x^{(\alpha)}$ by enforcing **monotonic nesting**, meaning that higher severities include *all* transformations from lower values of $\alpha$. Transformations are applied sequentially and are cumulative:

- $\alpha = 0$: no corruption (the original image is returned).

- $\alpha \geq 1$: **channel shuffle** (permute the RGB channels, e.g., $[R, G, B] \rightarrow [G, B, R]$), producing a consistent appearance shift while preserving spatial structure.

- $\alpha \geq 2$: **structured mark**: overlay a thin diagonal line across the image, introducing a global structured artifact.

- $\alpha \geq 3$: **global noise + additional line**: add Gaussian noise to all pixels (with clipping to [0,255]) and draw a second diagonal line, increasing visual clutter and reducing the signal-to-noise ratio.

- $\alpha \geq 4$: **strong structural and photometric shift**: rotate the image by 270°, invert pixel intensities (image negative), and apply another round of Gaussian noise.

# G. Additional Results

## G.1. Additional Results on Fairness-Aware Dataset Distillation

In addition to the EOD metric introduced in Section 3.2, we extend this definition to the maximum and mean EOD gaps as follows. Let $\mathcal{A} = \{a_1, a_2, \ldots, a_K\}$ denote the set of demographic groups, and let $A$ be the protected attribute. A sample $(x, y)$ is assigned to group $a_i$ if $A = a_i$. Let $\hat{Y} = f(X)$ be the model prediction, and define $p_y^{\mathrm{EOD}}(a) := \Pr(\hat{Y} = y \mid Y = y, \ A = a)$. We quantify disparities using the maximum and average EOD gaps:

$$\mathrm{EOD}_M = \sup_{y \in \mathcal{Y}, \, a_i, a_j \in \mathcal{A}} \left| p_y^{\mathrm{EOD}}(a_i) - p_y^{\mathrm{EOD}}(a_j) \right|, \qquad \mathrm{EOD}_A = \mathbb{E}_{y \in \mathcal{Y}} \left[ \sup_{a_i, a_j \in \mathcal{A}} \left| p_y^{\mathrm{EOD}}(a_i) - p_y^{\mathrm{EOD}}(a_j) \right| \right].$$

In the following, we present these metrics in the experimental setting described in Section 5.1, along with their statistical significance evaluated over 10 independent runs.

*Table 8.* Accuracy (%) on biased benchmarks. Each entry reports Acc ↑ (mean ± std) over 10 independent runs.

| Dataset | IPC | DC | | | DM | | | CAFE | | | IDC | | |
|---|---|---|---|---|---|---|---|---|---|---|---|---|---|
| | | Vanilla | FairDD | COBRA | Vanilla | FairDD | COBRA | Vanilla | FairDD | COBRA | Vanilla | FairDD | COBRA |
| UTKFace | 10 | 60.43 ± 3.95 | 61.02 ± 3.64 | 62.83 ± 3.09 | 66.79 ± 1.12 | 67.50 ± 0.80 | 70.65 ± 0.68 | 64.64 ± 0.64 | 65.01 ± 1.30 | 66.08 ± 0.89 | 55.88 ± 0.65 | 60.48 ± 0.85 | 55.60 ± 0.38 |
| | 50 | 71.02 ± 0.67 | 70.86 ± 1.09 | 72.31 ± 0.81 | 72.31 ± 0.50 | 74.18 ± 0.42 | 74.83 ± 0.63 | 69.85 ± 0.99 | 71.72 ± 0.71 | 71.50 ± 0.69 | 58.33 ± 0.37 | 55.97 ± 0.51 | 57.85 ± 0.56 |
| | 100 | 70.78 ± 1.11 | 70.04 ± 1.68 | 72.33 ± 1.43 | 72.86 ± 0.28 | 76.01 ± 0.34 | 75.88 ± 0.53 | 72.64 ± 0.61 | 73.75 ± 1.10 | 74.79 ± 0.35 | 38.35 ± 1.48 | 37.30 ± 0.88 | 37.42 ± 1.45 |
| C-MNIST (FG) | 10 | 72.01 ± 1.07 | 91.40 ± 0.26 | 91.65 ± 0.32 | 26.20 ± 0.49 | 94.02 ± 0.19 | 94.09 ± 0.18 | 22.06 ± 0.30 | 92.77 ± 0.16 | 92.10 ± 0.25 | 28.18 ± 1.43 | 92.75 ± 0.20 | 91.97 ± 0.27 |
| | 50 | 88.50 ± 1.17 | 92.38 ± 0.21 | 95.30 ± 0.10 | 59.57 ± 1.54 | 96.09 ± 0.07 | 96.89 ± 0.11 | 47.10 ± 0.93 | 95.91 ± 0.16 | 96.12 ± 0.11 | 42.54 ± 1.06 | 94.16 ± 0.07 | 93.54 ± 0.09 |
| | 100 | 85.60 ± 1.20 | 92.83 ± 0.37 | 95.12 ± 0.37 | 71.14 ± 1.60 | 96.97 ± 0.10 | 97.74 ± 0.07 | 66.52 ± 0.94 | 96.65 ± 0.05 | 97.10 ± 0.11 | 34.30 ± 0.86 | 93.96 ± 0.21 | 92.56 ± 0.28 |
| C-MNIST (BG) | 10 | 67.63 ± 2.03 | 91.44 ± 0.36 | 91.27 ± 0.21 | 20.32 ± 1.11 | 94.67 ± 0.07 | 94.45 ± 0.18 | 18.53 ± 0.90 | 79.74 ± 1.96 | 89.56 ± 0.54 | 19.30 ± 0.35 | 91.63 ± 0.21 | 90.99 ± 0.13 |
| | 50 | 88.98 ± 0.71 | 92.66 ± 0.28 | 93.50 ± 0.19 | 48.81 ± 1.60 | 96.40 ± 0.05 | 96.76 ± 0.14 | 47.59 ± 2.00 | 81.18 ± 1.25 | 93.43 ± 0.20 | 36.78 ± 1.55 | 93.92 ± 0.17 | 92.27 ± 0.25 |
| | 100 | 87.27 ± 1.06 | 92.44 ± 0.45 | 92.56 ± 0.19 | 70.27 ± 1.53 | 97.24 ± 0.05 | 97.50 ± 0.06 | 66.07 ± 1.35 | 80.05 ± 1.00 | 94.94 ± 0.18 | 40.83 ± 0.68 | 93.17 ± 0.22 | 80.46 ± 1.81 |
| C-FMNIST (FG) | 10 | 61.96 ± 1.09 | 75.21 ± 0.57 | 77.00 ± 0.22 | 33.85 ± 0.83 | 76.57 ± 0.29 | 77.07 ± 0.21 | 27.91 ± 0.63 | 73.66 ± 0.36 | 74.56 ± 0.35 | 41.77 ± 0.57 | 75.82 ± 0.27 | 75.89 ± 0.23 |
| | 50 | 65.74 ± 0.35 | 75.34 ± 0.12 | 74.52 ± 0.24 | 49.90 ± 0.59 | 81.44 ± 0.18 | 82.87 ± 0.12 | 48.15 ± 1.20 | 80.33 ± 0.08 | 81.07 ± 0.18 | 57.00 ± 0.51 | 77.08 ± 0.13 | 79.55 ± 0.26 |
| | 100 | 61.23 ± 1.26 | 68.57 ± 1.25 | 77.15 ± 0.61 | 58.09 ± 0.34 | 82.81 ± 0.22 | 84.48 ± 0.24 | 54.72 ± 1.15 | 82.02 ± 0.15 | 83.41 ± 0.12 | 57.95 ± 0.55 | 78.20 ± 0.14 | 77.41 ± 0.24 |
| C-FMNIST (BG) | 10 | 49.32 ± 0.77 | 69.04 ± 0.38 | 70.43 ± 0.40 | 22.05 ± 0.57 | 70.91 ± 0.48 | 70.62 ± 0.18 | 23.06 ± 0.87 | 59.35 ± 0.37 | 63.52 ± 0.48 | 33.46 ± 1.01 | 67.78 ± 0.31 | 67.48 ± 0.46 |
| | 50 | 61.59 ± 0.75 | 76.06 ± 0.18 | 73.70 ± 0.22 | 36.70 ± 0.82 | 78.53 ± 0.20 | 79.83 ± 0.20 | 36.99 ± 0.54 | 66.36 ± 0.65 | 70.63 ± 0.38 | 41.12 ± 0.87 | 72.62 ± 0.30 | 71.66 ± 0.39 |
| | 100 | 63.64 ± 0.39 | 70.69 ± 0.49 | 74.98 ± 0.21 | 46.17 ± 0.78 | 79.84 ± 0.17 | 81.40 ± 0.13 | 42.77 ± 0.66 | 67.78 ± 0.35 | 74.20 ± 0.40 | 41.40 ± 0.99 | 71.79 ± 0.63 | 70.13 ± 0.62 |
| CIFAR10-S | 10 | 36.66 ± 0.64 | 40.70 ± 0.41 | 40.87 ± 0.49 | 37.25 ± 0.31 | 43.94 ± 0.62 | 44.50 ± 0.56 | 35.77 ± 0.54 | 36.00 ± 0.88 | 39.49 ± 0.46 | 34.64 ± 0.17 | 40.10 ± 0.45 | 40.75 ± 0.14 |
| | 50 | 39.46 ± 0.98 | 46.16 ± 0.44 | 46.63 ± 0.56 | 45.26 ± 0.60 | 57.73 ± 0.35 | 57.89 ± 0.24 | 43.47 ± 0.76 | 44.47 ± 0.48 | 54.13 ± 0.34 | 39.43 ± 0.33 | 51.24 ± 0.83 | 50.13 ± 0.65 |
| | 100 | 41.84 ± 0.56 | 49.23 ± 0.49 | 50.45 ± 0.74 | 45.44 ± 0.46 | 61.20 ± 0.44 | 62.38 ± 0.52 | 43.24 ± 0.71 | 44.14 ± 0.75 | 58.02 ± 0.45 | 40.09 ± 0.42 | 51.14 ± 0.29 | 50.77 ± 0.41 |
| BFFHQ | 10 | 61.32 ± 0.97 | 63.78 ± 2.25 | 65.22 ± 2.68 | 65.12 ± 0.95 | 68.42 ± 1.00 | 69.47 ± 0.78 | 63.53 ± 0.82 | 63.80 ± 1.27 | 65.34 ± 0.89 | 62.06 ± 0.41 | 64.14 ± 0.36 | 66.26 ± 0.36 |
| | 50 | 64.45 ± 0.94 | 69.83 ± 0.60 | 71.58 ± 0.88 | 62.93 ± 0.58 | 72.62 ± 1.04 | 72.45 ± 0.91 | 65.52 ± 0.31 | 70.12 ± 0.54 | 70.65 ± 0.79 | 59.60 ± 0.71 | 63.60 ± 0.70 | 65.90 ± 0.81 |
| | 100 | 65.55 ± 0.71 | 69.02 ± 1.37 | 71.07 ± 2.42 | 65.77 ± 0.39 | 74.42 ± 0.72 | 74.25 ± 0.26 | 65.47 ± 0.60 | 71.95 ± 0.85 | 74.68 ± 0.59 | 51.14 ± 1.85 | 50.22 ± 1.61 | 50.88 ± 0.61 |

*Table 9.* Maximum Equalized Odds Difference (%) on biased benchmarks. Each entry reports $\mathrm{EOD}_M$ ↓ (mean ± std) over 10 independent runs.

| Dataset | IPC | DC | | | DM | | | CAFE | | | IDC | | |
|---|---|---|---|---|---|---|---|---|---|---|---|---|---|
| | | Vanilla | FairDD | COBRA | Vanilla | FairDD | COBRA | Vanilla | FairDD | COBRA | Vanilla | FairDD | COBRA |
| UTKFace | 10 | 50.83 ± 4.45 | 45.83 ± 6.07 | 36.67 ± 6.60 | 54.17 ± 4.14 | 40.17 ± 5.27 | 38.33 ± 1.89 | 66.83 ± 3.44 | 41.17 ± 3.93 | 40.70 ± 4.12 | 61.40 ± 1.36 | 49.20 ± 3.25 | 45.20 ± 2.79 |
| | 50 | 51.83 ± 2.03 | 35.67 ± 5.34 | 29.00 ± 2.38 | 53.50 ± 1.89 | 38.83 ± 1.77 | 35.00 ± 2.00 | 51.67 ± 2.21 | 43.33 ± 0.94 | 40.30 ± 2.15 | 52.20 ± 2.71 | 46.20 ± 1.83 | 38.80 ± 2.56 |
| | 100 | 47.83 ± 3.44 | 26.50 ± 1.98 | 26.67 ± 3.04 | 48.83 ± 1.67 | 31.00 ± 1.29 | 32.33 ± 2.36 | 46.67 ± 2.43 | 41.00 ± 1.63 | 36.83 ± 2.03 | 18.40 ± 5.46 | 13.60 ± 5.39 | 12.14 ± 4.74 |
| C-MNIST (FG) | 10 | 99.49 ± 0.51 | 22.88 ± 3.27 | 21.01 ± 2.89 | 100.00 ± 0.00 | 15.02 ± 2.86 | 14.74 ± 3.99 | 100.00 ± 0.00 | 11.73 ± 2.04 | 18.10 ± 2.94 | 100.00 ± 0.00 | 13.26 ± 1.83 | 18.78 ± 1.89 |
| | 50 | 64.86 ± 9.31 | 26.44 ± 3.30 | 10.39 ± 1.04 | 100.00 ± 0.00 | 10.57 ± 2.48 | 8.10 ± 0.98 | 100.00 ± 0.00 | 11.38 ± 0.96 | 10.01 ± 1.21 | 100.00 ± 0.00 | 17.55 ± 1.00 | 17.35 ± 1.58 |
| | 100 | 83.09 ± 8.78 | 22.44 ± 6.63 | 17.39 ± 3.61 | 100.00 ± 0.00 | 8.12 ± 0.68 | 7.62 ± 0.88 | 100.00 ± 0.00 | 9.75 ± 0.82 | 8.95 ± 0.76 | 100.00 ± 0.00 | 14.67 ± 2.28 | 18.40 ± 2.18 |
| C-MNIST (BG) | 10 | 99.67 ± 0.47 | 20.79 ± 5.00 | 17.51 ± 2.51 | 100.00 ± 0.00 | 15.12 ± 2.82 | 12.99 ± 2.42 | 100.00 ± 0.00 | 91.14 ± 3.29 | 17.29 ± 3.62 | 100.00 ± 0.00 | 18.13 ± 4.25 | 13.13 ± 1.93 |
| | 50 | 58.91 ± 9.19 | 19.77 ± 4.79 | 27.26 ± 3.88 | 100.00 ± 0.00 | 8.45 ± 0.73 | 7.46 ± 0.78 | 100.00 ± 0.00 | 90.39 ± 8.15 | 17.18 ± 2.28 | 100.00 ± 0.00 | 14.90 ± 1.66 | 17.10 ± 2.72 |
| | 100 | 94.55 ± 3.32 | 56.09 ± 14.35 | 30.09 ± 1.84 | 100.00 ± 0.00 | 8.29 ± 0.89 | 7.58 ± 1.09 | 100.00 ± 0.00 | 97.45 ± 1.51 | 13.29 ± 0.87 | 100.00 ± 0.00 | 14.82 ± 2.14 | 81.79 ± 8.62 |
| C-FMNIST (FG) | 10 | 99.00 ± 0.58 | 53.83 ± 5.87 | 37.00 ± 5.16 | 100.00 ± 0.00 | 29.50 ± 1.26 | 25.83 ± 3.53 | 100.00 ± 0.00 | 45.50 ± 6.29 | 25.50 ± 3.77 | 100.00 ± 0.00 | 47.20 ± 6.82 | 36.00 ± 3.41 |
| | 50 | 100.00 ± 0.00 | 47.50 ± 5.71 | 23.00 ± 4.04 | 100.00 ± 0.00 | 25.67 ± 2.21 | 21.00 ± 3.00 | 100.00 ± 0.00 | 21.50 ± 2.36 | 25.40 ± 2.11 | 100.00 ± 0.00 | 35.00 ± 3.85 | 29.00 ± 2.00 |
| | 100 | 100.00 ± 0.00 | 74.83 ± 13.97 | 36.83 ± 3.85 | 100.00 ± 0.00 | 26.33 ± 2.36 | 24.17 ± 1.34 | 100.00 ± 0.00 | 21.83 ± 3.58 | 19.50 ± 2.97 | 100.00 ± 0.00 | 40.60 ± 6.83 | 31.80 ± 3.97 |
| C-FMNIST (BG) | 10 | 100.00 ± 0.00 | 61.00 ± 5.42 | 34.50 ± 3.40 | 100.00 ± 0.00 | 44.67 ± 3.64 | 33.80 ± 1.72 | 100.00 ± 0.00 | 90.17 ± 2.34 | 46.70 ± 9.92 | 100.00 ± 0.00 | 47.20 ± 1.60 | 30.40 ± 3.50 |
| | 50 | 100.00 ± 0.00 | 50.00 ± 7.81 | 27.17 ± 3.24 | 100.00 ± 0.00 | 25.17 ± 2.27 | 21.40 ± 1.36 | 100.00 ± 0.00 | 90.00 ± 4.51 | 53.00 ± 5.66 | 100.00 ± 0.00 | 36.60 ± 4.84 | 34.00 ± 2.00 |
| | 100 | 99.00 ± 0.00 | 50.67 ± 5.02 | 36.67 ± 4.07 | 100.00 ± 0.00 | 28.00 ± 2.94 | 22.40 ± 3.26 | 100.00 ± 0.00 | 95.17 ± 1.77 | 65.40 ± 8.01 | 100.00 ± 0.00 | 46.00 ± 4.20 | 38.80 ± 1.17 |
| CIFAR10-S | 10 | 47.12 ± 3.12 | 26.30 ± 3.64 | 17.83 ± 1.99 | 56.25 ± 1.70 | 25.58 ± 2.34 | 20.18 ± 1.42 | 62.27 ± 3.58 | 39.97 ± 4.64 | 30.48 ± 1.90 | 54.60 ± 1.68 | 28.14 ± 2.70 | 22.42 ± 2.48 |
| | 50 | 71.85 ± 2.45 | 35.65 ± 1.73 | 26.18 ± 2.40 | 73.22 ± 2.07 | 25.40 ± 2.31 | 16.70 ± 2.65 | 61.93 ± 1.41 | 36.10 ± 3.48 | 30.28 ± 2.59 | 63.54 ± 1.01 | 37.68 ± 1.13 | 25.12 ± 0.99 |
| | 100 | 69.37 ± 2.50 | 41.43 ± 3.21 | 21.15 ± 3.67 | 82.87 ± 0.94 | 25.17 ± 3.65 | 9.37 ± 1.00 | 73.33 ± 1.98 | 27.73 ± 3.65 | 18.70 ± 4.58 | 60.64 ± 2.69 | 30.80 ± 1.18 | 22.44 ± 1.57 |
| BFFHQ | 10 | 48.80 ± 3.89 | 43.80 ± 6.07 | 32.87 ± 6.87 | 44.80 ± 1.53 | 22.13 ± 2.47 | 15.73 ± 1.24 | 37.20 ± 3.27 | 22.33 ± 2.71 | 19.56 ± 3.22 | 45.20 ± 2.26 | 33.60 ± 1.43 | 22.96 ± 2.68 |
| | 50 | 52.40 ± 3.16 | 53.53 ± 3.89 | 40.73 ± 5.21 | 60.27 ± 0.88 | 24.13 ± 3.21 | 15.13 ± 2.26 | 62.87 ± 2.30 | 38.40 ± 3.12 | 16.64 ± 3.01 | 35.76 ± 1.82 | 33.52 ± 3.64 | 23.68 ± 0.47 |
| | 100 | 59.60 ± 1.53 | 56.93 ± 4.54 | 44.07 ± 4.66 | 63.47 ± 1.53 | 22.53 ± 1.49 | 7.87 ± 1.36 | 65.27 ± 1.52 | 34.80 ± 2.41 | 15.36 ± 3.01 | 7.12 ± 4.26 | 5.60 ± 2.32 | 5.28 ± 3.50 |

*Table 10.* Mean Equalized Odds Difference (%) on biased benchmarks. Each entry reports $\text{EOD}_A \downarrow$ (mean $\pm$ std) over 10 independent runs.

| Dataset | IPC | DC | | | DM | | | CAFE | | | IDC | | |
|---|---|---|---|---|---|---|---|---|---|---|---|---|---|
| | | Vanilla | FairDD | COBRA | Vanilla | FairDD | COBRA | Vanilla | FairDD | COBRA | Vanilla | FairDD | COBRA |
| UTKFace | 10 | 32.61 ± 3.87 | 22.39 ± 1.81 | 20.44 ± 1.24 | 33.39 ± 2.53 | 22.78 ± 3.25 | 20.78 ± 1.73 | 45.95 ± 2.77 | 22.67 ± 2.90 | 21.07 ± 1.83 | 35.53 ± 1.41 | 27.00 ± 1.47 | 21.06 ± 1.60 |
| | 50 | 33.78 ± 1.52 | 23.11 ± 1.76 | 19.78 ± 1.73 | 33.44 ± 1.24 | 23.00 ± 0.61 | 20.61 ± 0.65 | 33.83 ± 1.86 | 24.72 ± 1.43 | 23.93 ± 1.26 | 24.13 ± 1.66 | 21.47 ± 1.20 | 18.80 ± 0.88 |
| | 100 | 34.22 ± 1.83 | 20.05 ± 1.75 | 21.39 ± 2.81 | 32.66 ± 1.33 | 21.28 ± 1.57 | 19.16 ± 1.40 | 31.50 ± 2.04 | 24.61 ± 1.14 | 23.67 ± 1.67 | 10.40 ± 1.65 | 7.73 ± 2.67 | 7.05 ± 1.16 |
| C-MNIST (FG) | 10 | 63.64 ± 2.54 | 10.34 ± 0.72 | 8.96 ± 0.43 | 99.97 ± 0.05 | 7.26 ± 0.50 | 6.71 ± 0.18 | 99.98 ± 0.04 | 7.49 ± 0.65 | 9.72 ± 0.57 | 99.82 ± 0.12 | 7.38 ± 0.31 | 8.36 ± 0.48 |
| | 50 | 26.68 ± 2.46 | 10.51 ± 0.42 | 5.96 ± 0.13 | 87.51 ± 2.50 | 5.62 ± 0.46 | 4.74 ± 0.27 | 96.52 ± 0.98 | 6.99 ± 0.33 | 5.88 ± 0.47 | 99.07 ± 0.41 | 7.21 ± 0.28 | 7.76 ± 0.31 |
| | 100 | 43.60 ± 5.62 | 10.70 ± 1.32 | 8.05 ± 0.93 | 73.06 ± 4.69 | 4.62 ± 0.15 | 4.12 ± 0.16 | 76.49 ± 2.24 | 5.84 ± 0.40 | 5.19 ± 0.48 | 99.69 ± 0.16 | 7.39 ± 0.38 | 8.95 ± 0.62 |
| C-MNIST (BG) | 10 | 71.07 ± 3.71 | 10.07 ± 1.40 | 8.80 ± 1.02 | 99.63 ± 0.07 | 7.51 ± 0.28 | 7.04 ± 0.69 | 99.68 ± 0.07 | 47.84 ± 5.42 | 10.10 ± 0.59 | 99.86 ± 0.10 | 9.66 ± 0.62 | 8.15 ± 0.46 |
| | 50 | 23.91 ± 1.75 | 9.63 ± 1.36 | 8.39 ± 0.59 | 98.92 ± 0.55 | 5.31 ± 0.23 | 5.10 ± 0.34 | 97.96 ± 0.65 | 51.41 ± 4.10 | 8.74 ± 0.62 | 98.35 ± 0.87 | 7.50 ± 0.41 | 9.05 ± 0.42 |
| | 100 | 36.44 ± 3.02 | 14.25 ± 2.02 | 11.91 ± 1.35 | 76.36 ± 3.44 | 4.81 ± 0.18 | 4.74 ± 0.37 | 82.91 ± 2.77 | 56.28 ± 1.90 | 7.67 ± 0.40 | 98.20 ± 0.50 | 9.05 ± 0.85 | 41.93 ± 5.20 |
| C-FMNIST (FG) | 10 | 74.10 ± 2.12 | 26.02 ± 2.35 | 17.43 ± 0.56 | 99.20 ± 0.06 | 17.25 ± 0.48 | 15.93 ± 0.98 | 99.00 ± 0.18 | 22.85 ± 1.22 | 16.64 ± 1.02 | 98.68 ± 0.37 | 21.34 ± 1.16 | 16.90 ± 1.16 |
| | 50 | 76.23 ± 1.69 | 21.05 ± 1.26 | 11.12 ± 1.01 | 96.43 ± 0.82 | 14.82 ± 0.85 | 12.65 ± 0.80 | 95.82 ± 1.36 | 12.23 ± 0.41 | 14.49 ± 0.72 | 89.92 ± 2.28 | 18.06 ± 0.39 | 14.38 ± 0.90 |
| | 100 | 84.82 ± 3.17 | 27.23 ± 2.52 | 15.87 ± 0.72 | 87.83 ± 0.57 | 13.43 ± 0.70 | 11.70 ± 0.56 | 88.83 ± 0.91 | 11.45 ± 0.95 | 11.76 ± 0.69 | 89.82 ± 1.06 | 17.52 ± 1.10 | 16.80 ± 0.84 |
| C-FMNIST (BG) | 10 | 90.10 ± 0.37 | 32.92 ± 1.62 | 21.38 ± 1.35 | 99.50 ± 0.08 | 24.05 ± 0.40 | 21.48 ± 1.23 | 99.57 ± 0.05 | 51.73 ± 2.24 | 25.80 ± 1.87 | 99.30 ± 0.13 | 27.88 ± 1.16 | 19.00 ± 0.86 |
| | 50 | 72.08 ± 0.84 | 24.93 ± 2.14 | 16.40 ± 0.77 | 99.30 ± 0.15 | 15.80 ± 1.07 | 13.70 ± 0.94 | 99.50 ± 0.14 | 50.02 ± 2.22 | 28.20 ± 2.26 | 99.34 ± 0.23 | 19.90 ± 1.26 | 20.02 ± 1.37 |
| | 100 | 64.03 ± 1.82 | 23.15 ± 1.35 | 19.08 ± 1.83 | 98.17 ± 0.93 | 15.17 ± 0.36 | 13.48 ± 1.01 | 98.10 ± 0.40 | 49.08 ± 1.55 | 32.83 ± 2.46 | 97.84 ± 0.48 | 25.02 ± 1.57 | 24.52 ± 1.05 |
| CIFAR10-S | 10 | 27.58 ± 2.44 | 9.30 ± 0.59 | 6.77 ± 0.75 | 39.10 ± 0.43 | 9.18 ± 0.29 | 9.56 ± 0.62 | 41.96 ± 1.76 | 21.75 ± 2.99 | 14.32 ± 1.41 | 29.22 ± 0.82 | 8.78 ± 0.79 | 7.81 ± 0.40 |
| | 50 | 42.78 ± 2.43 | 8.05 ± 0.60 | 7.54 ± 0.57 | 52.32 ± 1.24 | 7.96 ± 1.02 | 5.97 ± 0.44 | 49.39 ± 1.44 | 17.60 ± 1.45 | 11.78 ± 1.55 | 42.14 ± 0.49 | 10.91 ± 0.68 | 9.65 ± 0.84 |
| | 100 | 44.08 ± 2.38 | 11.77 ± 0.90 | 7.10 ± 0.66 | 59.95 ± 1.02 | 8.15 ± 1.08 | 4.37 ± 0.25 | 57.74 ± 1.26 | 15.94 ± 1.06 | 7.74 ± 0.97 | 39.10 ± 0.92 | 9.62 ± 0.32 | 8.46 ± 0.25 |
| BFFHQ | 10 | 38.17 ± 3.97 | 33.50 ± 3.35 | 21.23 ± 4.82 | 36.43 ± 2.37 | 15.83 ± 2.27 | 9.27 ± 0.91 | 33.40 ± 2.20 | 12.27 ± 1.20 | 12.96 ± 2.14 | 40.28 ± 2.16 | 21.72 ± 1.14 | 18.12 ± 1.04 |
| | 50 | 43.83 ± 2.82 | 40.00 ± 2.53 | 29.97 ± 2.63 | 50.80 ± 1.24 | 18.50 ± 1.95 | 8.90 ± 1.13 | 51.17 ± 1.35 | 26.43 ± 1.36 | 11.34 ± 2.32 | 34.00 ± 1.78 | 30.96 ± 2.12 | 22.60 ± 0.54 |
| | 100 | 48.77 ± 0.93 | 44.03 ± 1.59 | 32.67 ± 1.77 | 53.33 ± 0.66 | 18.17 ± 0.64 | 5.63 ± 0.87 | 53.40 ± 1.11 | 27.83 ± 0.95 | 11.84 ± 2.35 | 5.24 ± 3.55 | 4.28 ± 1.61 | 3.36 ± 1.69 |

## G.2. Further Details on Ablation for the Worst-Case Residual Bound

**Empirical verification protocol.** We empirically verify Theorem 4.1 by comparing, for each class $y$, how well two different targets summarize class-conditional subgroup statistics in a worst-case sense. For a fixed dataset and a fixed distilled set, we reconstruct a *converged* model by training a fresh network from scratch on the distilled images within a fixed budget. We then freeze this trained network and compute class-conditional subgroup statistics $\{\Phi_{a|y}\}_{a \in \mathcal{A}}$ from the *real* training set, where $a$ indexes the sensitive attribute. For each class $y$ and each subgroup $a$, we estimate $\Phi_{a|y}$. We use objective-dependent definitions of $\Phi$: for **DC**, $\Phi_{a|y}$ is the flattened parameter-gradient vector of the cross-entropy loss computed on a real minibatch from subgroup $a$ (matching gradient-based distillation statistics). For **DM**, $\Phi_{a|y}$ is the subgroup mean feature embedding $\Phi_{a|y} = \mu_{a|y} = \mathbb{E}[\text{embed}(x) \mid a, y]$, computed by passing the real examples through the network's embedding function and averaging in feature space.

**Targets, residuals, and theorem conditions.** Given $\{\Phi_{a|y}\}_{a \in \mathcal{A}}$ for a class $y$, we form the two targets compared in Theorem 4.1: the *vanilla (mixture) target* $m_y^{\text{van}} = \sum_a \pi_{a|y} \Phi_{a|y}$, where $\pi_{a|y}$ is the empirical subgroup proportion within class $y$, and the *COBRA barycenter target* $m_y^*$ defined as the uniform barycenter over subgroups. Under the squared Euclidean geometry used in our verification, the uniform barycenter reduces to the $m_y^* = \frac{1}{|\mathcal{A}_y|} \sum_a \Phi_{a|y}$. We then compute the worst-case residual for each target as $\max_a \|\Phi_{a|y} - m_y\|_2^2$ and record whether COBRA reduces this maximum residual relative to the vanilla target, i.e., $\max_a \|\Phi_{a|y} - m_y^*\|_2^2 \leq \max_a \|\Phi_{a|y} - m_y^{\text{van}}\|_2^2$. To assess when the theorem's sufficient condition applies, we additionally compute the dot-product condition: letting $a^+$ be the subgroup attaining the maximum COBRA residual, $\Delta_{a^+|y}^C = \Phi_{a^+|y} - m_y^*$, and $s_y = m_y^{\text{van}} - m_y^*$, we evaluate $\langle \Delta_{a^+|y}^C, s_y \rangle$ and mark the condition as holding when it is non-positive. We repeat this evaluation for every class (skipping classes with only one observed subgroup) and aggregate the results across datasets (BFFHQ, CIFAR10-S), distilled set sizes (IPC $\in \{10, 50\}$), and report both class-averaged and worst-class $\text{MaxRes}_{\text{COBRA}}$ used in Table 5.

## G.3. Ablation Study with Small IPC

Table 11 demonstrates that in the low-IPC setting, dataset distillation severely worsens fairness: for both datasets (CIFAR10-S, BFFHQ) and both distillation backbones (DC, DM), greater imbalance-induced difficulty consistently raises EOD, reflecting stronger bias amplification. In contrast, COBRA consistently achieves the lowest (or tied-lowest) EOD across nearly all settings while maintaining comparable accuracy. On CIFAR10-S, COBRA reduces EOD sharply at IPC = 1 (e.g., $18.0 \rightarrow 6.2$ for DC) and remains the best as IPC increases. On BFFHQ, COBRA improves upon both Vanilla and FairDD for DC at all IPCs, and for DM, it matches or surpasses FairDD, with particularly large gains at IPC = 3 and 5 (e.g., $11.1 \rightarrow 7.8$) without sacrificing accuracy. Overall, these results indicate that COBRA is robust in the low-IPC regime, providing consistent fairness improvements across distillation methods while preserving predictive performance.

*Table 11.* Sensitivity to low IPC. Entries report max EOD↓ with accuracy↑ in parentheses.

| Dataset | IPC | DC | | | DM | | |
|---|---|---|---|---|---|---|---|
| | | Vanilla | FairDD | COBRA | Vanilla | FairDD | COBRA |
| CIFAR10-S | 1 | 18.0 (25.8) | 8.4 (26.2) | 6.2 (25.8) | 17.2 (24.4) | 14.6 (24.8) | 4.9 (23.6) |
| | 3 | 37.4 (32.3) | 25.5 (33.3) | 20.3 (33.2) | 47.6 (31.1) | 29.8 (34.5) | 23.4 (34.5) |
| | 5 | 41.4 (32.7) | 30.5 (34.6) | 21.2 (34.8) | 51.0 (34.0) | 28.4 (38.1) | 27.5 (39.5) |
| BFFHQ | 1 | 32.2 (59.6) | 23.5 (61.7) | 15.1 (60.6) | 27.6 (61.3) | 19.9 (62.1) | 19.0 (61.9) |
| | 3 | 34.6 (60.1) | 29.0 (60.6) | 16.5 (58.0) | 30.4 (60.9) | 11.1 (61.9) | 7.8 (58.3) |
| | 5 | 38.9 (61.4) | 35.4 (60.9) | 21.8 (61.5) | 43.2 (62.4) | 10.5 (63.9) | 8.6 (64.4) |

# H. Statistical Significance for Group Imbalance and Group Separation Results

In Tables 12 and 13, we report the statistically annotated versions of the original results from Table 2, discussed in Section 5.2. Specifically, the reported values include the standard deviation over 10 runs, augmenting the initial findings with measures of statistical variability and thereby offering a more comprehensive assessment of the robustness and reliability of the observed patterns.

*Table 12.* Accuracy (Acc) as SKEW varies (top) on CIFAR10-S with fixed group GAP and IPC =50, and as group GAP varies (bottom) at fixed SKEW =0.8. Results are reported for DC/DM/IDC distilled datasets (Vanilla, FairDD, COBRA), together with the FULL baseline.

| Method | Value | DC | | | DM | | | IDC | | | FULL |
|---|---|---|---|---|---|---|---|---|---|---|---|
| | | Vanilla | FairDD | COBRA | Vanilla | FairDD | COBRA | Vanilla | FairDD | COBRA | |
| SKEW | 0.60 | 44.58 ± 0.15 | 45.67 ± 0.32 | 46.44 ± 0.23 | 49.23 ± 0.59 | 51.56 ± 0.33 | 54.00 ± 0.44 | 43.00 ± 0.48 | 46.72 ± 0.61 | 46.39 ± 0.77 | 75.15 ± 0.12 |
| | 0.65 | 43.72 ± 0.58 | 45.78 ± 0.39 | 46.17 ± 0.21 | 48.03 ± 0.23 | 51.38 ± 0.37 | 52.76 ± 0.37 | 42.08 ± 0.53 | 47.06 ± 0.52 | 46.20 ± 0.54 | 74.65 ± 0.15 |
| | 0.70 | 41.39 ± 0.30 | 45.43 ± 0.27 | 45.49 ± 0.52 | 46.30 ± 0.39 | 50.98 ± 0.47 | 52.92 ± 0.39 | 39.77 ± 0.37 | 46.13 ± 0.44 | 46.07 ± 0.50 | 73.99 ± 0.28 |
| | 0.75 | 39.24 ± 0.44 | 44.76 ± 0.53 | 45.37 ± 0.37 | 44.27 ± 0.47 | 50.58 ± 0.45 | 52.49 ± 0.36 | 37.29 ± 0.43 | 46.75 ± 0.40 | 46.35 ± 0.78 | 72.59 ± 0.07 |
| | 0.80 | 37.63 ± 0.47 | 44.48 ± 0.42 | 45.22 ± 0.25 | 42.27 ± 0.55 | 50.14 ± 0.38 | 52.60 ± 0.26 | 36.07 ± 0.10 | 47.18 ± 0.74 | 46.15 ± 0.23 | 70.67 ± 0.30 |
| | 0.85 | 35.91 ± 0.29 | 43.43 ± 0.31 | 44.60 ± 0.30 | 39.24 ± 0.37 | 49.61 ± 0.29 | 51.60 ± 0.31 | 34.62 ± 0.37 | 46.17 ± 0.47 | 45.96 ± 0.95 | 68.44 ± 0.11 |
| GAP | 1 | 37.14 ± 0.22 | 41.85 ± 0.86 | 40.25 ± 1.00 | 40.56 ± 0.59 | 44.67 ± 0.37 | 44.52 ± 0.22 | 37.24 ± 0.18 | 42.36 ± 0.59 | 41.05 ± 0.59 | 82.56 ± 0.02 |
| | 2 | 34.46 ± 0.17 | 39.98 ± 0.68 | 40.20 ± 0.99 | 35.68 ± 0.21 | 45.76 ± 0.51 | 44.89 ± 0.50 | 36.27 ± 0.45 | 41.91 ± 0.52 | 42.01 ± 0.85 | 79.60 ± 0.07 |
| | 3 | 30.82 ± 0.36 | 36.75 ± 0.71 | 35.25 ± 1.05 | 30.13 ± 0.25 | 40.71 ± 0.58 | 41.19 ± 0.60 | 34.21 ± 0.64 | 39.61 ± 0.65 | 39.36 ± 0.26 | 65.93 ± 0.67 |
| | 4 | 27.43 ± 0.37 | 33.29 ± 0.90 | 31.21 ± 2.07 | 29.21 ± 0.27 | 37.70 ± 0.32 | 37.34 ± 0.30 | 31.13 ± 0.33 | 35.94 ± 0.56 | 36.23 ± 0.35 | 62.88 ± 0.03 |

*Table 13.* Equal opportunity difference (EOD) as SKEW varies (top) on CIFAR10-S with fixed group GAP and IPC =50, and as group GAP varies (bottom) at fixed SKEW =0.8. Results are reported for DC/DM/IDC distilled datasets (Vanilla, FairDD, COBRA), together with the FULL baseline.

| Method | Value | DC | | | DM | | | IDC | | | FULL |
|---|---|---|---|---|---|---|---|---|---|---|---|
| | | Vanilla | FairDD | COBRA | Vanilla | FairDD | COBRA | Vanilla | FairDD | COBRA | |
| SKEW | 0.60 | 32.24 ± 2.85 | 13.64 ± 2.15 | 10.16 ± 1.33 | 27.05 ± 2.62 | 9.13 ± 1.65 | 8.53 ± 1.74 | 34.50 ± 2.33 | 10.23 ± 1.15 | 8.08 ± 1.64 | 14.13 ± 0.69 |
| | 0.65 | 45.10 ± 2.81 | 14.48 ± 2.16 | 11.28 ± 1.42 | 33.13 ± 1.71 | 11.65 ± 1.53 | 8.99 ± 2.72 | 44.87 ± 0.63 | 12.48 ± 2.77 | 8.40 ± 0.73 | 19.13 ± 0.52 |
| | 0.70 | 57.29 ± 1.88 | 15.00 ± 2.09 | 11.33 ± 1.21 | 43.53 ± 1.93 | 12.90 ± 2.17 | 10.83 ± 2.01 | 55.53 ± 3.18 | 12.23 ± 1.58 | 7.43 ± 1.88 | 26.27 ± 1.06 |
| | 0.75 | 65.81 ± 1.53 | 19.55 ± 3.05 | 15.51 ± 2.59 | 54.59 ± 0.67 | 17.00 ± 1.53 | 11.19 ± 2.09 | 65.23 ± 0.62 | 13.73 ± 2.02 | 7.82 ± 1.67 | 33.90 ± 0.64 |
| | 0.80 | 70.43 ± 1.78 | 26.05 ± 3.77 | 14.70 ± 2.32 | 61.94 ± 1.06 | 22.94 ± 2.44 | 12.59 ± 1.32 | 70.63 ± 2.07 | 16.63 ± 1.51 | 7.65 ± 1.63 | 42.73 ± 0.74 |
| | 0.85 | 76.38 ± 1.27 | 32.96 ± 2.38 | 17.68 ± 2.86 | 78.54 ± 1.37 | 24.84 ± 2.07 | 13.27 ± 2.59 | 73.13 ± 1.45 | 14.65 ± 0.80 | 7.65 ± 2.20 | 52.60 ± 0.22 |
| GAP | 1 | 30.93 ± 1.17 | 10.15 ± 1.57 | 7.98 ± 1.73 | 46.13 ± 4.80 | 16.12 ± 1.25 | 8.42 ± 1.19 | 41.87 ± 1.15 | 7.98 ± 0.45 | 4.63 ± 0.40 | 27.30 ± 0.20 |
| | 2 | 50.65 ± 2.05 | 27.12 ± 3.54 | 11.32 ± 0.87 | 62.23 ± 1.52 | 12.53 ± 2.73 | 9.10 ± 0.91 | 53.77 ± 0.93 | 10.10 ± 1.52 | 9.90 ± 1.74 | 39.50 ± 2.40 |
| | 3 | 56.75 ± 2.05 | 27.75 ± 4.60 | 11.73 ± 1.79 | 72.40 ± 0.51 | 19.73 ± 3.06 | 17.28 ± 3.34 | 56.53 ± 2.22 | 17.80 ± 1.07 | 12.98 ± 1.33 | 54.85 ± 2.95 |
| | 4 | 61.40 ± 1.34 | 38.22 ± 7.23 | 16.38 ± 4.53 | 74.90 ± 0.43 | 19.70 ± 0.81 | 19.52 ± 2.49 | 57.43 ± 0.76 | 24.80 ± 0.63 | 18.95 ± 1.26 | 59.00 ± 0.80 |

# I. Computational Overhead of COBRA

We provide a detailed analysis of the per-iteration wall-clock time and peak GPU memory incurred by COBRA relative to Vanilla DD. Vanilla DD computes one real-data distillation target per class. COBRA replaces this monolithic target with a group-wise barycenter, requiring $G = |\mathcal{A}|$ separate passes (one per demographic group), whose outputs are then averaged into a uniform barycenter. Here, $d$ denotes the dimensionality of the group-wise quantity being averaged to form the barycenter. The barycenter computation itself is only an elementwise average over the $G$ group-specific vectors, with a

*Table 14.* Per-iteration runtime (ms) and peak GPU memory (MB) on C-FMNIST (BG) as the number of groups $G$ increases. The first row is Vanilla DD; the remaining rows are COBRA. "Slowdown" and "Mem. Mult." denote runtime and memory relative to Vanilla DD. Runtime is reported as mean $\pm$ std over 120 measured iterations after 10 warm-up iterations.

| | DC | | | | DM | | | |
|---|---|---|---|---|---|---|---|---|
| G | Time (ms) | Slowdown | Mem (MB) | Mem. Mult. | Time (ms) | Slowdown | Mem (MB) | Mem. Mult. |
| – | $83.8 \pm 13.6$ | baseline | 1,566 | baseline | $17.4 \pm 4.1$ | baseline | 1,176 | baseline |
| 2 | $131.0 \pm 21.1$ | $1.56\times$ | 1,855 | $1.18\times$ | $21.7 \pm 5.3$ | $1.25\times$ | 1,029 | $0.87\times$ |
| 4 | $190.6 \pm 26.0$ | $2.28\times$ | 1,857 | $1.19\times$ | $28.8 \pm 4.8$ | $1.66\times$ | 1,061 | $0.90\times$ |
| 6 | $244.5 \pm 19.7$ | $2.92\times$ | 1,864 | $1.19\times$ | $36.7 \pm 9.6$ | $2.11\times$ | 1,091 | $0.93\times$ |
| 8 | $302.7 \pm 22.1$ | $3.61\times$ | 1,862 | $1.19\times$ | $44.2 \pm 9.4$ | $2.55\times$ | 1,122 | $0.95\times$ |
| 10 | $359.4 \pm 24.2$ | $4.29\times$ | 1,868 | $1.19\times$ | $52.4 \pm 12.2$ | $3.02\times$ | 1,151 | $0.98\times$ |

*Table 15.* Per-iteration runtime (ms) and peak GPU memory (MB) at each dataset's natural group count. We report the change from Vanilla DD to COBRA, together with the corresponding runtime and memory multipliers. Runtime is reported as mean $\pm$ std over 120 measured iterations after 10 warm-up iterations.

| | | DC | | | | DM | | | |
|---|---|---|---|---|---|---|---|---|---|
| Dataset | G | Time (ms) | Slowdown | Mem. (MB) | Mem. Mult. | Time (ms) | Slowdown | Mem. (MB) | Mem. Mult. |
| CIFAR10-S | 2 | $83.3 \pm 13.2 \rightarrow 132.5 \pm 22.2$ | $1.59\times$ | $1,447 \rightarrow 1,737$ | $1.20\times$ | $16.7 \pm 0.1 \rightarrow 26.6 \pm 8.2$ | $1.59\times$ | $1,058 \rightarrow 1,033$ | $0.98\times$ |
| BFFHQ | 2 | $38.0 \pm 10.7 \rightarrow 52.7 \pm 8.3$ | $1.39\times$ | $3,146 \rightarrow 4,277$ | $1.36\times$ | $10.0 \pm 0.1 \rightarrow 13.0 \pm 4.5$ | $1.30\times$ | $2,068 \rightarrow 2,075$ | $1.00\times$ |
| C-FMNIST | 10 | $83.8 \pm 13.6 \rightarrow 359.4 \pm 24.2$ | $4.29\times$ | $1,566 \rightarrow 1,868$ | $1.19\times$ | $17.4 \pm 4.1 \rightarrow 52.4 \pm 12.2$ | $3.02\times$ | $1,176 \rightarrow 1,151$ | $0.98\times$ |
| C-MNIST | 10 | $96.7 \pm 39.8 \rightarrow 361.6 \pm 40.3$ | $3.74\times$ | $1,566 \rightarrow 1,868$ | $1.19\times$ | $17.4 \pm 4.7 \rightarrow 55.8 \pm 16.5$ | $3.21\times$ | $1,176 \rightarrow 1,151$ | $0.98\times$ |

cost of $\mathcal{O}(Gd)$, and is therefore negligible in practice. The dominant cost comes from computing these group-wise quantities in the first place, which requires repeated forward/backward evaluations of the network.

All experiments are run on a single **NVIDIA H100 80 GB** GPU with an Intel Xeon Gold 6442Y CPU and 2 TB RAM. We use a `ConvNet` backbone with IPC = 10 and a real-data batch size of 256 per class. Timings are averaged over 120 measured iterations, after 10 warm-up iterations excluded from the statistics, and are measured with full CUDA synchronization (`torch.cuda.synchronize`) before and after each iteration. Peak GPU memory is tracked using `torch.cuda.max_memory_allocated`.

Table 14 reports both runtime and memory as the number of groups is increased on C-FMNIST (BG). Runtime grows predictably with $G$: for DC, the slowdown increases from $1.56\times$ at $G = 2$ to $4.29\times$ at $G = 10$, while for DM it increases from $1.25\times$ to $3.02\times$. This trend is expected since each group requires an additional real-data pass, and DC is more sensitive because it performs repeated backward passes through the retained real-data computation graph. In contrast, the memory overhead is modest. For DC, peak memory only rises from $1.18\times$ to $1.19\times$ of Vanilla DD across the entire sweep, while for DM it stays essentially unchanged and is even slightly lower than Vanilla DD in our measurements ($0.87\times$–$0.98\times$), consistent with the sequential execution of the group-wise forward passes. Even at $G = 10$, the absolute per-iteration times remain in the tens-to-hundreds of milliseconds range; for example, a full DC run of 2,400 iterations still completes in under 15 minutes on an H100.

Table 15 summarizes the same quantities at each dataset's natural group count. In the practically common two-group setting (e.g., BFFHQ and CIFAR10-S), the runtime overhead is moderate, with DC slowdown in the range of $1.39\times$–$1.59\times$ and DM slowdown in the range of $1.30\times$–$1.59\times$. Memory remains well controlled: DC uses only $1.20\times$–$1.36\times$ the memory of Vanilla DD, while DM stays nearly identical to Vanilla DD ($0.98\times$–$1.00\times$). Larger runtime overhead appears only in the intentionally extreme 10-group synthetic settings (C-FMNIST and C-MNIST), where DC reaches $4.29\times$ and $3.74\times$ slowdown and DM reaches $3.02\times$ and $3.21\times$, yet the memory increase is still small for DC ($1.19\times$) and negligible for DM ($0.98\times$).

Overall, COBRA introduces a predictable computational overhead because it replaces a single class-level real-data pass with $G$ group-wise passes. In the practically common two-group setting, this leads to a moderate runtime increase while keeping memory overhead low. In more extreme synthetic settings with many groups, the runtime cost becomes larger but remains tractable, and peak memory still changes only modestly. Given the substantial fairness gains in the main paper, we view this additional cost as well justified.

## J. Ablation Study on Robustness to Group Label Noise

COBRA depends on per-sample demographic group labels to construct its cross-group barycentric target. In real-world scenarios, however, these labels are often imperfect because they may be self-reported, inferred from proxies, or affected by annotation noise. We therefore examine how robust COBRA is to corrupted group annotations during distillation. Concretely, we randomly choose a fraction $\rho \in \{10\%, 15\%, 20\%, 50\%\}$ of the training instances and replace each chosen demographic label with a uniformly sampled *incorrect* group index. All other elements of the training procedure are left unchanged.

Table 16 presents the results for DM with IPC= 10. Overall, they show that COBRA's performance deteriorates smoothly as group labels become noisier. On C-MNIST (BG), the method is especially stable as accuracy stays close to 94% across all corruption rates, and EOD increases only moderately from 12.99% under clean labels to 17.92% at 50% noise. This indicates that, for controlled benchmarks with a clear signal-to-noise separation, the barycentric objective remains effective even when a sizable fraction of sensitive attributes is corrupted.

A comparable, though somewhat weaker, trend appears on CIFAR10-S. As the noise level grows, accuracy decreases slightly from 44.50% to 41.17%, and EOD increases from 20.18% to 37.03%. Crucially, even with noisy labels, COBRA remains substantially less biased than Vanilla DD, whose EOD reaches 56.25%, while also sustaining a consistent accuracy edge. This suggests that COBRA's fairness improvements do not rely on perfectly accurate group supervision. On BFFHQ, the effect of corrupted group labels is more pronounced. Both accuracy and fairness degrade progressively with increasing $\rho$, and EOD rises from 15.73% with clean labels to 38.48% at 50% noise. Nonetheless, COBRA continues to deliver fairness gains over Vanilla DD at all corruption levels. However, the accuracy benefit seen in the clean scenario largely vanishes once noise is introduced, implying that real-world datasets with richer subgroup structures are more vulnerable to inaccuracies in protected attribute labels.

Overall, these findings suggest that COBRA exhibits reasonable robustness to moderate levels of demographic label noise. Although corrupting group annotations weakens the fairness signal that underlies the barycentric target, the approach consistently offers fairness advantages over fairness-agnostic distillation and often maintains competitive predictive accuracy. Such robustness is particularly valuable in practical applications, where sensitive attributes are commonly noisy, incomplete, or only approximately observed.

*Table 16.* Robustness to noisy sensitive-group labels under DM with IPC = 10. We report accuracy and EOD. The first two column blocks present clean-label results for Vanilla DD and COBRA, while the remaining show COBRA performance with noisy group labels.

| Dataset | Vanilla DD | | COBRA Clean | | COBRA + 10% noise | | COBRA + 15% noise | | COBRA + 20% noise | | COBRA + 50% noise | |
|---|---|---|---|---|---|---|---|---|---|---|---|---|
| | Acc | EOD | Acc | EOD | Acc | EOD | Acc | EOD | Acc | EOD | Acc | EOD |
| C-MNIST (BG) | $20.32 \pm 1.11$ | $100.0 \pm 0.0$ | $\mathbf{94.45 \pm 0.18}$ | $\mathbf{12.99 \pm 2.42}$ | $94.07 \pm 0.13$ | $13.33 \pm 1.77$ | $93.87 \pm 0.14$ | $15.05 \pm 2.46$ | $94.63 \pm 0.09$ | $14.70 \pm 1.10$ | $94.05 \pm 0.20$ | $17.92 \pm 2.91$ |
| CIFAR10-S | $37.25 \pm 0.31$ | $56.25 \pm 1.70$ | $\mathbf{44.50 \pm 0.56}$ | $\mathbf{20.18 \pm 1.42}$ | $42.61 \pm 0.48$ | $28.60 \pm 2.41$ | $42.39 \pm 0.45$ | $31.00 \pm 1.24$ | $42.96 \pm 0.22$ | $31.48 \pm 3.24$ | $41.17 \pm 0.43$ | $37.03 \pm 2.70$ |
| BFFHQ | $65.12 \pm 0.95$ | $44.80 \pm 1.53$ | $\mathbf{69.47 \pm 0.78}$ | $\mathbf{15.73 \pm 1.24}$ | $63.65 \pm 2.22$ | $25.90 \pm 2.92$ | $64.13 \pm 0.88$ | $26.10 \pm 2.12$ | $64.58 \pm 1.24$ | $28.08 \pm 2.72$ | $62.10 \pm 0.84$ | $38.48 \pm 2.69$ |

## K. Ablation Study on Partially Available Group Labels

In many practical settings, however, such demographic group labels are only available for a subset of the training data. To evaluate this regime, we simulate partial group annotation by retaining the sensitive labels for only a fraction $\eta \in \{75\%, 50\%, 25\%, 10\%, 5\%\}$ of the training set and masking the rest. To recover the missing group information, we adopt a simple pseudo-labeling procedure based on unsupervised clustering. We first flatten the input images and run K-means over the full training set using only visual features, without access to group annotations. After clustering, each cluster is assigned a sensitive-group identity by majority vote over the samples in that cluster whose labels remain known. This cluster-level label is then propagated to all unlabeled samples in the same cluster, yielding pseudo-group labels for the missing portion of the dataset. COBRA is subsequently trained using the combination of observed and imputed group labels.

Table 17 reports the results under DM with IPC= 10. Overall, reducing the fraction of known sensitive labels degrades both accuracy and fairness, as expected; however, COBRA consistently remains less biased than Vanilla DD across all datasets and supervision levels. On C-MNIST (FG), the method is fairly robust with 50%–75% known labels; performance stays close to the clean-label COBRA result, and even at 10% or 5% known labels, COBRA still yields a much lower EOD than Vanilla DD. A stronger dependence on group supervision is observed on C-FMNIST (BG). As the known-label ratio decreases, accuracy drops and EOD rises, especially below 25%. Still, even in the severe 10% and 5% settings, COBRA remains clearly better than Vanilla DD in terms of fairness, showing that the pseudo-labeling strategy continues to provide a

useful group signal.

On CIFAR10-S and BFFHQ, partial labels lead to a larger gap from the clean-label COBRA setting, indicating that pseudo-group recovery is less reliable on these more complex datasets. Nevertheless, COBRA retains a consistent fairness advantage over Vanilla DD throughout, including under the most challenging $10\%$ and $5\%$ label-availability regimes. Taken together, these results show that COBRA remains effective when sensitive attributes are only partially observed. Although unsupervised label imputation does not fully recover the clean-label setting, it preserves a meaningful portion of COBRA's fairness benefit, even under very limited group supervision.

*Table 17.* Robustness to partially available sensitive-group labels under DM with IPC= 10. The first two column groups show clean-label results for Vanilla DD and COBRA; the remaining groups show COBRA when only a fraction of sensitive labels is known.

| Dataset | Vanilla DD | | COBRA | | COBRA + 75% known | | COBRA + 50% known | | COBRA + 25% known | | COBRA + 10% known | | COBRA + 5% known | |
|---|---|---|---|---|---|---|---|---|---|---|---|---|---|---|
| | Acc | EOD | Acc | EOD | Acc | EOD | Acc | EOD | Acc | EOD | Acc | EOD | Acc | EOD |
| C-MNIST (FG) | $26.20 \pm 0.49$ | $100.0 \pm 0.0$ | $\mathbf{94.09 \pm 0.18}$ | $\mathbf{14.74 \pm 3.99}$ | $93.76 \pm 0.14$ | $18.23 \pm 2.51$ | $93.86 \pm 0.41$ | $20.54 \pm 7.55$ | $93.52 \pm 0.19$ | $30.43 \pm 6.97$ | $92.37 \pm 0.40$ | $48.27 \pm 5.96$ | $92.69 \pm 0.20$ | $54.68 \pm 2.91$ |
| C-FMNIST (BG) | $22.05 \pm 0.57$ | $100.0 \pm 0.0$ | $\mathbf{77.07 \pm 0.21}$ | $\mathbf{33.80 \pm 1.72}$ | $70.62 \pm 0.50$ | $38.50 \pm 2.06$ | $69.39 \pm 0.46$ | $42.00 \pm 1.22$ | $68.18 \pm 0.42$ | $60.50 \pm 0.50$ | $67.91 \pm 0.39$ | $73.50 \pm 2.18$ | $67.27 \pm 0.61$ | $81.25 \pm 2.77$ |
| BFFHQ | $65.12 \pm 0.95$ | $44.80 \pm 1.53$ | $\mathbf{69.47 \pm 0.78}$ | $\mathbf{15.73 \pm 1.24}$ | $62.13 \pm 1.23$ | $34.60 \pm 2.20$ | $63.53 \pm 0.80$ | $39.30 \pm 4.83$ | $63.48 \pm 0.29$ | $35.50 \pm 0.77$ | $64.15 \pm 0.75$ | $38.80 \pm 3.34$ | $66.20 \pm 0.94$ | $41.60 \pm 2.47$ |
| CIFAR10-S | $37.25 \pm 0.31$ | $56.25 \pm 1.70$ | $\mathbf{44.50 \pm 0.56}$ | $\mathbf{20.18 \pm 1.42}$ | $41.09 \pm 0.42$ | $43.45 \pm 2.35$ | $38.24 \pm 0.49$ | $47.90 \pm 2.10$ | $39.14 \pm 0.58$ | $49.13 \pm 1.86$ | $37.59 \pm 0.44$ | $53.60 \pm 0.76$ | $38.14 \pm 0.43$ | $52.93 \pm 0.75$ |

# L. Ablation on Standard Fairness Interventions

We next examine whether the bias induced by dataset distillation can be mitigated by inserting standard fairness interventions into an otherwise vanilla pipeline. We study three intervention points. **Representation-level fairness (RF)** uses a fair feature extractor before distillation. **Data-level fairness (DF)** uses subgroup-aware loss reweighting during distillation. **Model-level fairness (MF)** keeps the distilled set unchanged and applies a fair training objective only when fitting the downstream model.

Table 18 shows a clear overall pattern. Interventions applied before or after distillation are generally unreliable, whereas reweighting in distillation is the strongest conventional baseline. This behavior is consistent with our main claim that the dominant source of bias lies in the aggregation target used during distillation, rather than solely in the upstream representation or the downstream learner.

The pre-distillation baseline RF is often insufficient. Even when the extractor is trained with a fairness-aware objective, vanilla distillation still aggregates subgroup statistics according to their empirical frequencies. As a result, the target can remain skewed toward the majority structure. This is especially visible on C-MNIST (BG) with DM, where RF leaves EOD at $100.0\%$ and yields almost no change relative to Vanilla DD. A similar pattern appears on CIFAR10-S with DM, where RF is slightly worse than Vanilla DD in fairness despite a modest gain in accuracy.

The post-distillation baseline MF is also inconsistent. Once the synthetic dataset has discarded subgroup-specific information, a downstream fairness objective can only rebalance the limited support that remains. It cannot reconstruct the minority structure that was not preserved during condensation. This limitation is again clear on C-MNIST (BG), where MF produces an EOD of $100.0\%$ under both DM and DC, and on BFFHQ, where it improves fairness only at the cost of a substantial drop in accuracy.

Among the standard interventions, DF is the most competitive. Reweighting directly counteracts subgroup imbalance during optimization and therefore delivers large fairness gains on CIFAR10-S and C-MNIST (BG). However, its improvements are not uniform across settings. One DC configuration on C-MNIST (BG) is unstable, and on BFFHQ the gains in EOD do not translate into the strongest fairness-accuracy balance. This suggests that correcting subgroup frequencies alone is not enough when subgroup statistics are also geometrically misaligned.

COBRA provides the most reliable overall trade-off because it changes the distillation target itself. Instead of reweighting samples or modifying only the downstream learner, it aligns the synthetic data to a subgroup-balanced barycentric target. This yields the lowest EOD on four of the six dataset-backbone pairs and remains competitive on the other two while preserving stronger accuracy. The clearest example is BFFHQ. Under DM, DF attains slightly lower EOD than COBRA, with $14.10\%$ versus $15.73\%$, but COBRA improves accuracy from $66.0\%$ to $69.5\%$. Under DC, MF obtains the best EOD, $30.20\%$, yet its accuracy drops to $57.45\%$, whereas COBRA reaches a comparable EOD of $32.87\%$ with much higher accuracy at $65.2\%$. Overall, these results indicate that fairness in dataset distillation cannot be treated as a simple add-on to the beginning or the end of the pipeline. The aggregation objective itself must be fairness-aware, which is precisely the role of COBRA.

*Table 18.* Comparison of COBRA with standard fairness interventions applied at different stages of the distillation pipeline for IPC = 10. Entries report EOD with accuracy in parentheses. RF denotes Fair Extractor, DF denotes Loss Reweighting, and MF denotes Fair Downstream. The best EOD in each row is shown in bold. Missing values correspond to unstable optimization.

| Dataset | Method | Vanilla DD | RF (Fair Extractor) | DF (Loss Reweighting) | MF (Fair Downstream) | COBRA |
|---|---|---|---|---|---|---|
| CIFAR10-S | DM | 56.25 (37.2) | 57.23 (38.9) | 25.63 (44.9) | 50.18 (33.1) | **20.18** (44.5) |
| | DC | 47.12 (36.7) | 29.40 (33.1) | 27.18 (33.1) | 40.75 (31.7) | **17.83** (40.9) |
| C-MNIST (BG) | DM | 100.00 (20.3) | 100.00 (20.9) | 13.53 (94.3) | 100.00 (29.0) | **12.99** (94.5) |
| | DC | 99.67 (67.6) | 31.99 (85.9) | - | 100.00 (31.8) | **17.51** (91.3) |
| BFFHQ | DM | 44.80 (65.1) | 43.50 (64.4) | **14.10** (66.0) | 37.90 (60.6) | 15.73 (69.5) |
| | DC | 48.80 (61.3) | 36.50 (60.6) | 39.00 (62.98) | **30.20** (57.45) | 32.87 (65.2) |

### L.1. Isolating Bias Amplification via Semantic-Preserving Separation

In our initial analysis of group separation (Appendix F and In Section 5.2), we found that pronounced representational divergence between subgroups coincides with a sharp rise in EOD. Nonetheless, to conclusively demonstrate that this bias amplification stems directly from the distillation aggregation mechanism rather than from a mere weakening of the underlying class-discriminative signal, we need to disentangle representational distance from overall task difficulty.

To achieve this, we design a *Semantic-Preserving GAP* experiment. Instead of applying destructive image-space corruptions, we artificially induce representational separation by adding a constant, group-specific orthogonal offset vector to the input space. Specifically, for a given image $x$ belonging to a minority subgroup $a$, the modified input becomes $x' = x + \gamma \cdot \mathbf{v}_a$, where $\mathbf{v}_a$ is a constant color-channel shift, and $\gamma \in \{1, 2, 3, 4\}$ acts as our separation multiplier (GAP).

Because $\mathbf{v}_a$ is a constant shift, it does not destroy the original semantic features required for class prediction. Consequently, a model trained on the FULL dataset can easily learn to ignore this offset. Table 19 presents the results of this controlled ablation on CIFAR10-S (IPC=10). This controlled setting reveals three key findings:

- **FULL Baseline Stability:** As the GAP increases, the FULL baseline maintains a highly stable accuracy ($\sim$72%) and a constant EOD ($\sim$51-52%). This confirms that the task itself does not become inherently harder or more biased as the geometric separation increases.

- **Vanilla DD Amplification:** Despite the task difficulty remaining constant, Vanilla DD suffers from bias amplification. As the groups move further apart, the synthetic target is pulled increasingly toward the unshifted majority, causing Vanilla DD's EOD to degrade (climbing to 68.2% at GAP=4) and its accuracy to drop.

- **COBRA Mitigation and Regularization:** COBRA successfully mitigates this geometric failure. By aligning the synthetic data to a subgroup-agnostic barycenter, COBRA not only prevents the amplification seen in Vanilla DD but acts as a strong fairness regularizer. It achieves an EOD of 13.3% to 29.3% while outperforming Vanilla DD in accuracy by nearly 10% across all GAP levels.

*Table 19.* Isolating the effect of representational separation (GAP) using a semantic-preserving orthogonal offset on CIFAR10-S (IPC=10). Unlike destructive corruptions, the FULL baseline performance remains stable, proving the EOD degradation in Vanilla DD is an artifact of the distillation process. COBRA suppresses this amplification and improves baseline fairness. Values are reported as EOD (Acc).

| Semantic GAP | FULL Baseline | Vanilla DM | DM + COBRA |
|---|---|---|---|
| 1 | 50.90 (72.15) | 61.00 (37.38) | **16.23** (45.1) |
| 2 | 52.10 (71.9) | 62.97 (35.0) | **13.30** (44.1) |
| 3 | 52.20 (72.4) | 63.73 (33.1) | **26.10** (42.8) |
| 4 | 50.60 (71.6) | 68.20 (32.9) | **29.33** (41.5) |

## M. Visualization Results

We provide visualizations at IPC = 10 on different datasets.

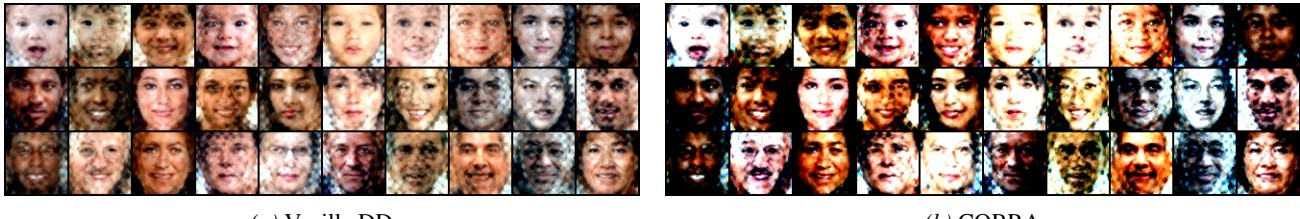

*(a)* Vanilla DD                                                                                         *(b)* COBRA

*Figure 5.* UTKFace (DM, IPC = 10). Panel (a) is Vanilla DD and panel (b) is COBRA.

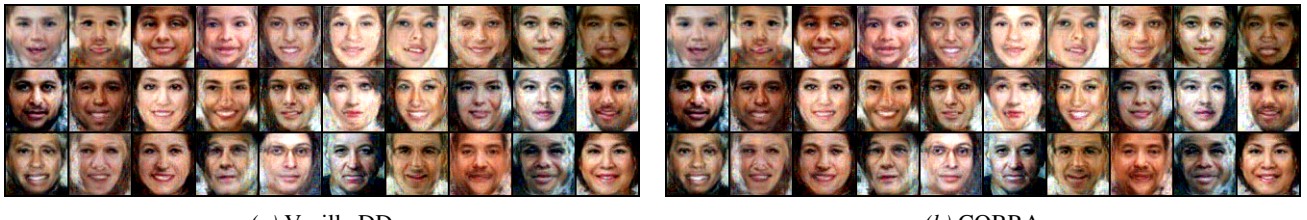

*(a)* Vanilla DD                                                                                         *(b)* COBRA

*Figure 6.* UTKFace (DC, IPC = 10). Panel (a) is Vanilla DD and panel (b) is COBRA.

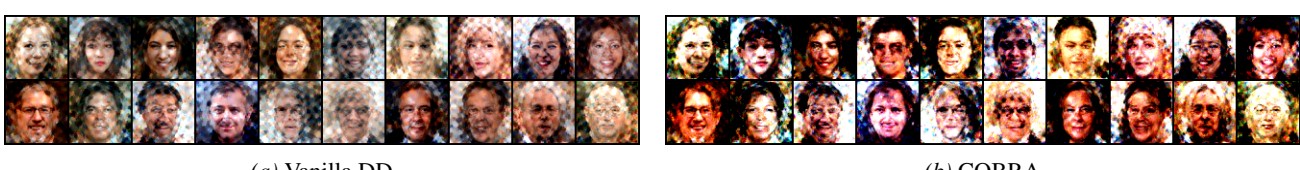

*(a)* Vanilla DD                                                                                         *(b)* COBRA

*Figure 7.* BFFHQ (DM, IPC = 10). Panel (a) is Vanilla DD and panel (b) is COBRA.

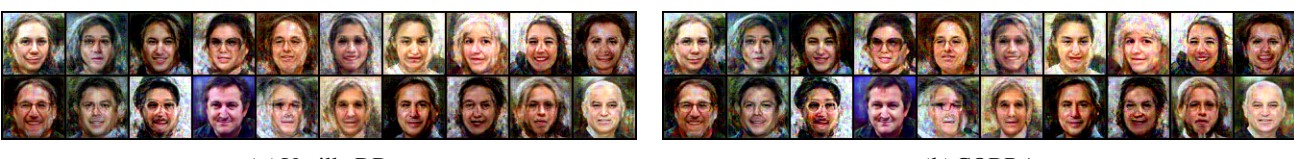

*(a)* Vanilla DD                                                                                         *(b)* COBRA

*Figure 8.* BFFHQ (DC, IPC = 10). Panel (a) is Vanilla DD and panel (b) is COBRA.

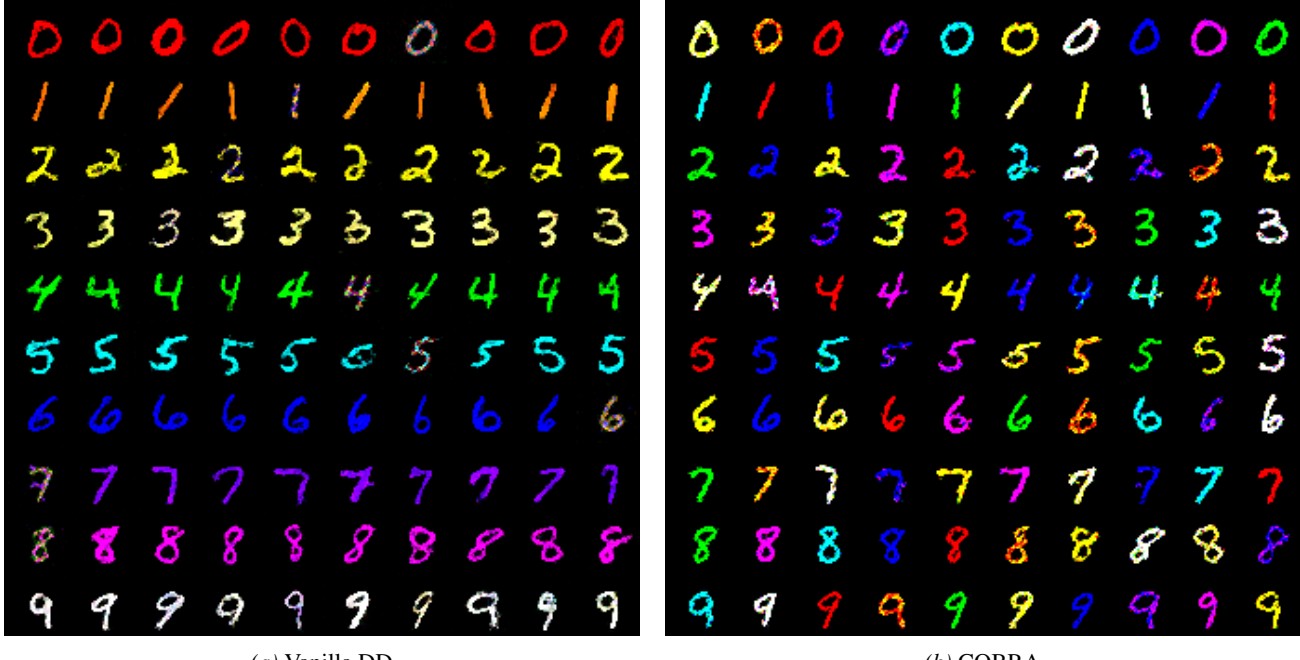

*(a)* Vanilla DD   *(b)* COBRA

*Figure 9.* Colored MNIST foreground (DM, IPC = 10). Panel (a) is Vanilla DD and panel (b) is COBRA.

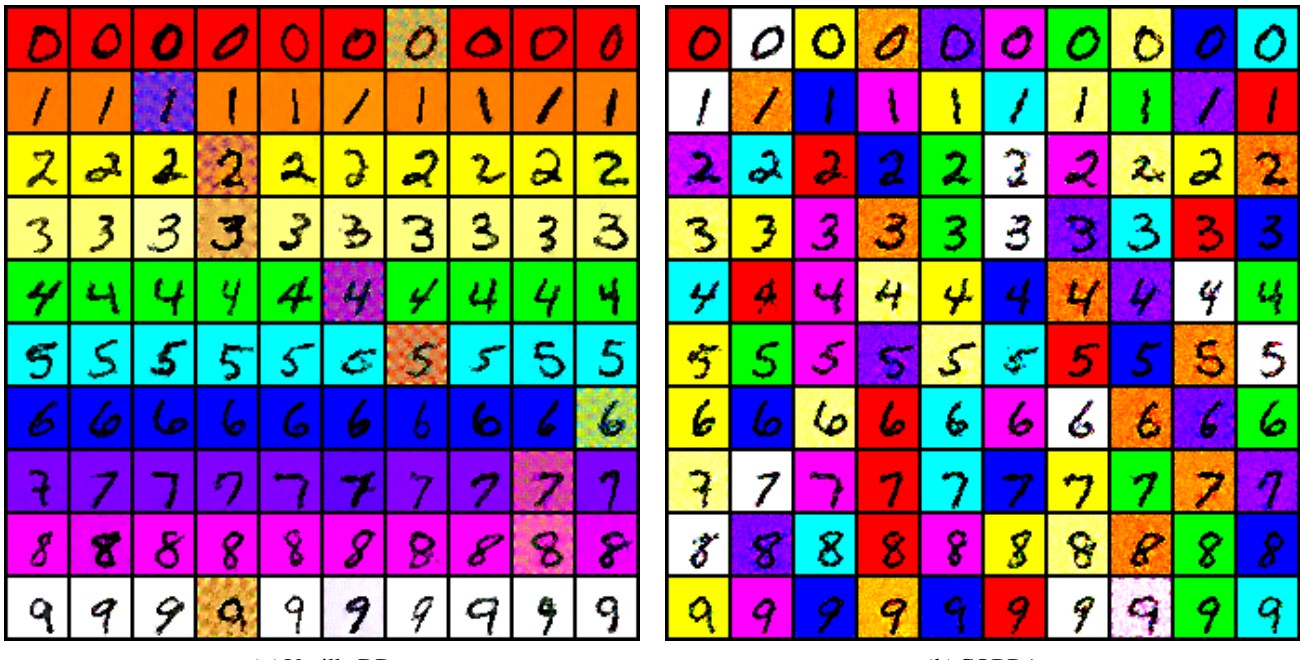

*(a)* Vanilla DD   *(b)* COBRA

*Figure 10.* Colored MNIST background (DM, IPC = 10). Panel (a) is Vanilla DD and panel (b) is COBRA.

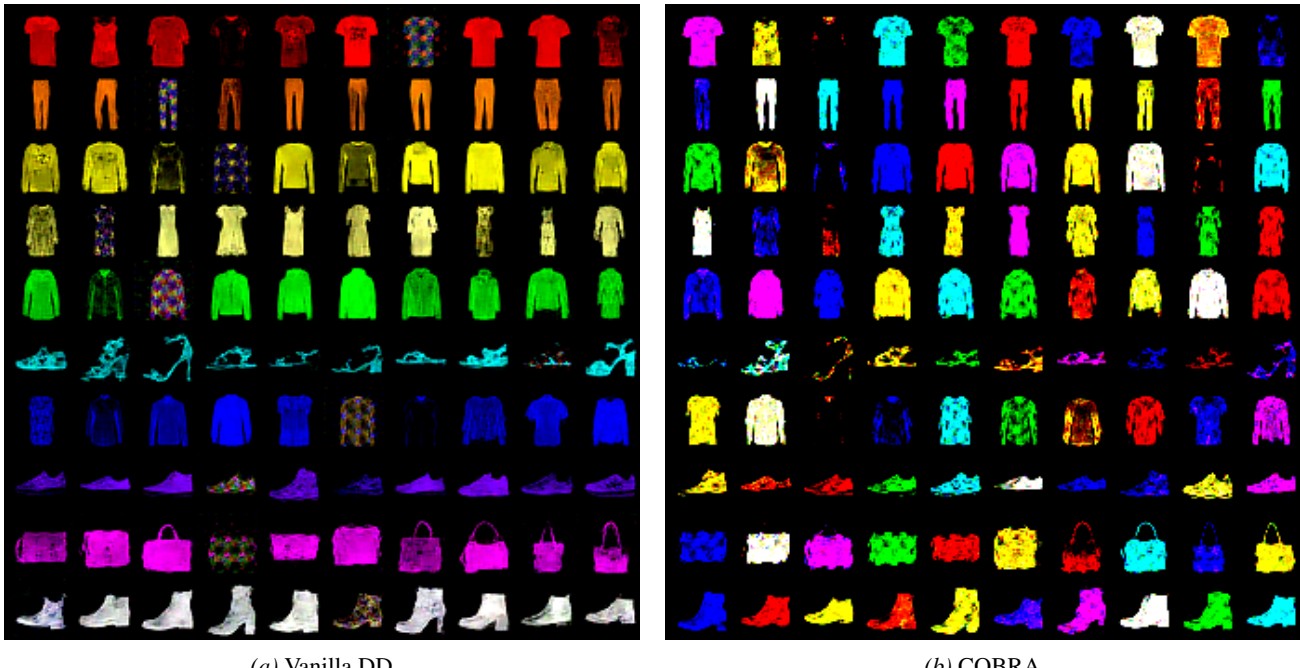

*(a)* Vanilla DD                    *(b)* COBRA

*Figure 11.* Colored FashionMNIST foreground (DM, IPC = 10). Panel (a) is Vanilla DD and panel (b) is COBRA.

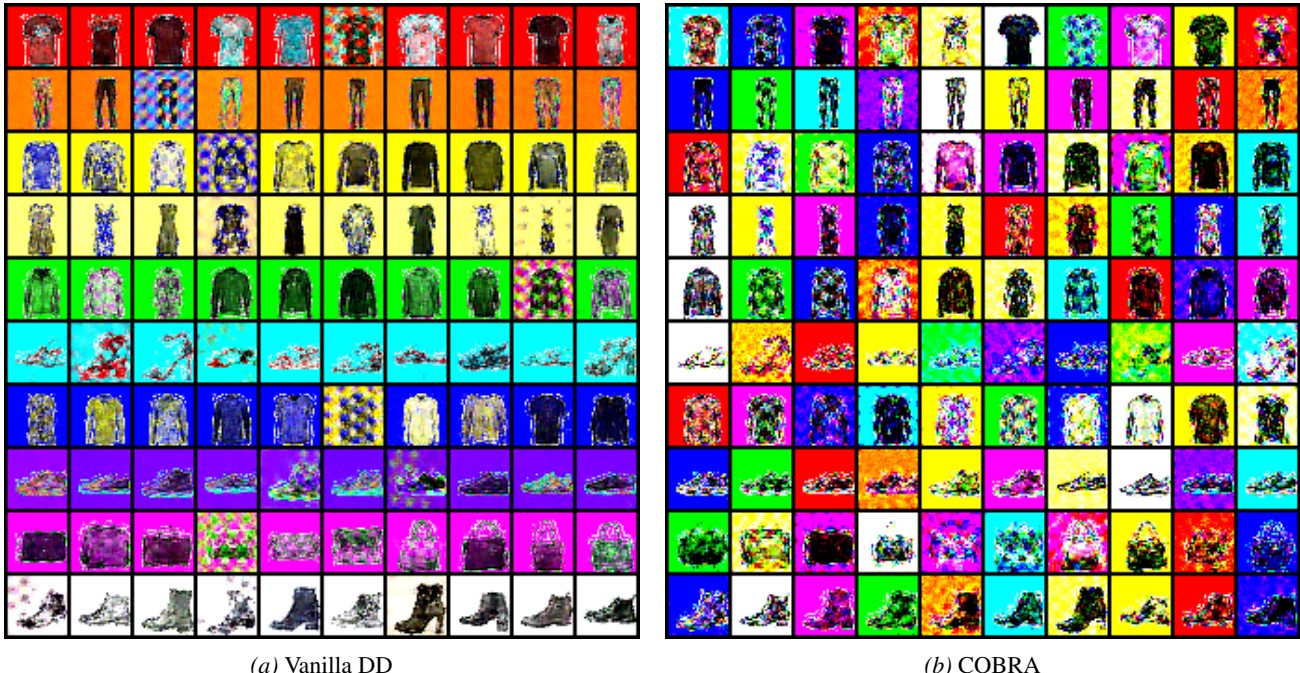

*(a)* Vanilla DD                    *(b)* COBRA

*Figure 12.* Colored FashionMNIST background (DM, IPC = 10). Panel (a) is Vanilla DD and panel (b) is COBRA.

