# OpenReview forum: "Fair Dataset Distillation via Cross-Group Barycenter Alignment"
_ICML.cc/2026/Conference — ICML 2026 regular_

### Official Review · Reviewer_pF7z · 2026-02-27

**Soundness:** 3
**Presentation:** 3
**Significance:** 3
**Originality:** 3
**Overall Recommendation:** 5
**Confidence:** 4

**Summary:**

This paper studies fairness degradation in dataset distillation (DD) and argues that bias amplification arises from the interaction between group imbalance and representational separation across subgroups, not from either factor in isolation. The authors derive an upper bound formalizing this interaction (Eq. 6) and propose COBRA, which replaces the vanilla aggregate distillation target with a uniform barycenter over per-class, per-group representation statistics. Under squared Q-norm discrepancy, this barycenter reduces to the simple uniform mean of subgroup statistics. Theorem 4.1 provides a sufficient condition under which COBRA does not increase the worst-case subgroup residual. Experiments cover C-MNIST, C-FMNIST, CIFAR10-S, UTKFace, and BFFHQ, evaluated against DC, DM, CAFE, and IDC backbones, with FairDD (Zhou et al., 2025) as the primary fairness competitor.

**Compliance With Llm Reviewing Policy:**

Affirmed.

**Final Justification:**

COBRA makes a clean contribution to an underexplored problem: the two-factor decomposition in Eq. (6) is a genuine conceptual advance over prior work, Theorem 4.1 and its proof are correct, and the cross-architecture generalization results are a strong practical validation. The rebuttal addressed all four of my weaknesses with actual experimental results: Table 2 variance is now provided confirming the 10-run protocol, MTT experiments were run and reported, the semantic-preserving GAP experiment with a stable FULL baseline cleanly isolates distillation-specific amplification, and concrete residual numbers in Theorem 4.1 violation cases show the sufficient condition is not binding in practice. Before camera-ready, the authors ought to qualify the plug-in framing for trajectory-based methods, add a caveat on the SKEW=0.60 overlap, and expand the limitations section to acknowledge EOD-only evaluation and group-label availability.

**Key Questions For Authors:**

From above weaknesses:

1. Can the authors provide variance estimates for all Table 2 entries (SKEW and GAP sweeps) consistent with the 10-run protocol used in Table 1?

2. Why is MTT excluded as a backbone? Does COBRA's representation-level target substitution apply to trajectory-based methods in principle?

3. In the GAP sweep, the FULL baseline EOD increases substantially with GAP, suggesting the task itself is getting harder in a way that independently increases fairness gaps. Can the authors demonstrate that the "abrupt degradation" in Section 5.2 reflects distillation-specific amplification rather than full-data baseline collapse? A separation mechanism that increases inter-group representational distance without degrading class-discriminative signal would help isolate the effect.

4. Can the authors construct a case where condition (9) fails and show whether COBRA still improves over vanilla in that setting?

**Limitations:**

The limitations discussion in the conclusion is thin. The authors mention future directions around noisy group annotations, multiple protected attributes, and distribution shift, but don't acknowledge that EOD is the only fairness metric evaluated, that group labels must be available at distillation time, or that the closed-form barycenter is specific to squared Q-norm. The restriction to binary or low-cardinality sensitive attributes is also not acknowledged, and UTKFace with four race groups is the only non-binary case in the experiments.

**Strengths And Weaknesses:**

Strengths:

1. The two-factor decomposition in Eq. (6) is clean and well-motivated. The controlled sweeps in Figure 2 isolate each effect independently, and this framing is a genuine conceptual advance over prior work such as Zhou et al. (2025), which treats imbalance alone as the driver of bias.

2. Theorem 4.1 and the supporting proof in Appendix C.3 are correct. Proposition C.1's strict convexity argument for uniqueness of the barycenter is rigorous, and Table 5 gives a credible empirical verification of the theorem's sufficient condition holding in practice across BFFHQ and CIFAR10-S.

3. The cross-architecture generalization experiment in Section 5.4 (Table 3) is a strong addition. The low cross-model variance in EOD and accuracy suggests COBRA's fairness gains aren't an artifact of ConvNet-specific inductive biases, which is an important practical claim to substantiate.

Weaknesses:

1. Table 2 is central to the causal claims about SKEW and GAP, but no standard deviations are reported anywhere in the paper for those results. Appendix G.1 covers only Table 1's datasets. For example, the COBRA vs. FairDD margin at SKEW=0.60, DM (8.53 vs. 8.99) is a difference of 0.46 EOD points, which is uninterpretable without any uncertainty estimate. The authors ought to report mean +/- std for all Table 2 entries with the same 10-run protocol used for Table 1, or at minimum add a supplementary table covering both sweeps.

2. MTT (Cazenavette et al., 2022) is discussed in Section 2 as a major DD paradigm but doesn't appear as a backbone anywhere in the experiments. This matters because MTT's trajectory-matching objective operates on a fundamentally different signal than DC or DM's representation-level statistics, and it's not obvious that COBRA's target substitution transfers to that setting. The authors ought to include MTT as a backbone on at least one benchmark, or explicitly argue why COBRA's analysis doesn't extend to trajectory-based methods and scope the claims accordingly.

3. The GAP construction in Appendix F seems to conflate representation separation with task difficulty. At GAP=4, the applied transformations include 270-degree rotation, pixel inversion, and two rounds of Gaussian noise. These don't just move groups apart in representation space; they make the task harder for everyone. In Table 2, the FULL baseline (no distillation) rises from EOD=0 at GAP=0 to EOD=59.0 at GAP=4, while accuracy drops from 86.8 to 62.9. So I'm not convinced the experiment isolates what it claims to isolate. The apparent "abrupt degradation" described in Section 5.2 is at least partly the full-data baseline collapsing, not distillation-specific amplification. Ideally, the separation mechanism would increase inter-group representational distance without degrading class-discriminative signal, for example by adding a fixed offset vector in feature space or using a style transfer that preserves semantic content. The current construction doesn't fully support the claim it's designed to demonstrate.

4. Condition (9) in Theorem 4.1 is a sufficient condition, but the paper doesn't discuss when it fails or how binding it is. Intuitively, the condition seems reasonable when the vanilla target is pulled away from the worst-case subgroup by majority pressure, but I'd like to understand whether there are configurations where it doesn't hold. Table 5 shows the condition holds empirically in the tested settings, but those settings were selected after the fact. Could the authors exhibit a synthetic two-subgroup case where condition (9) fails and report whether COBRA still reduces the worst-case residual empirically?


Summary:
The bias decomposition and COBRA framework are a clean contribution to an underexplored problem, and the empirical results are consistent across settings. The main gaps are the missing variance on Table 2, the absent MTT baseline, and the confounded GAP construction, each of which limits the strength of the causal claims. These all seem fixable in rebuttal. I'd encourage the authors to prioritize variance estimates for Table 2 and either add MTT or explicitly scope their theoretical claims to the backbone families they've analyzed.

---

> ### Author Rebuttal · Authors · 2026-03-30
>
> **W1, Q1:**
>
> - We include the full table with standard deviations here: [Link](https://anonymous.4open.science/r/figsstuff-7F69/standard_deviations.jpg).
> - All Table 2 results were obtained using the same 10-run protocol as Table 1.
> - At $\text{SKEW} = 0.60$ with DM, COBRA achieves $8.53 \pm 1.74\%$ EOD, while FairDD obtains $9.13 \pm 1.65\%$, showing similar variance but better average performance for COBRA.
>
> ------------
>
> **W2, Q2:**
>
> - We conducted additional experiments using MTT (Cazenavette et al., 2022); the results are available here: [Link](https://anonymous.4open.science/r/figsstuff-7F69/MTT.jpg).
> - These results suggest that **COBRA is not limited to DC or DM** and can also improve fairness under trajectory matching.
> - To adapt COBRA to MTT, we average checkpoints from demographic-specific replay buffers to construct a group-balanced barycentric trajectory target for the student network.
>
> ------------
>
> **W3, Q3:**
>
> - We ran a new **semantic-preserving GAP** experiment, with results available here: [Link](https://anonymous.4open.science/r/figsstuff-7F69/SemanticGAP.jpg).
> - The results show that the performance drop is caused by **distillation-specific bias amplification**, rather than by a harder full-data task.
> - **Method:** We used a constant group-specific orthogonal offset to increase inter-group distance while preserving semantic content.
> - **Baseline check:** The FULL baseline remained stable across GAP levels, showing that the task itself did not become harder.
> - **Vanilla DD degradation:** As GAP increases, Vanilla DD deteriorates.
> - **COBRA mitigation:** COBRA substantially reduces EOD across GAP levels while improving accuracy over Vanilla DD.
>
> ------------
>
> **W4, Q4:**
>
> - We agree that Eq. (9) in Theorem 4.1 is a **sufficient condition** and does not necessarily hold in all configurations.
> - However, even when it is **violated**, COBRA still **reduces the worst-case residual**.
> - Below we show representative cases:
>
> | Setting | Max residual (DM → COBRA) |
> |---|---:|
> | BFFHQ, IPC=10, Class 0 | 726.1 → 183.4 |
> | BFFHQ, IPC=50, Class 0 | 205.1 → 51.8 |
> | CIFAR10-S, IPC=10, Class 0 | 303.9 → 93.8 |
>
> --------
>
> **Limitation:**
>
> - We also analyzed **multi-group settings**, including UTKFace **(4 race categories)** and C-MNIST/C-FMNIST **(10 groups each)**. We agree that our current evaluation is still focused on low-to-moderate group cardinality, and we will make this limitation explicit in the paper.
> - We additionally evaluated the **sufficiency** fairness metric and observed the same overall trend (numbers below are sufficiency gaps in \%):
>
> | Dataset | DM | COBRA |
> |---|---:|---:|
> | CIFAR10-S | 52.16 | 9.19 |
> | C-MNIST (BG) | 100.0 | 6.42 |
> | BFFHQ | 31.27 | 7.16 |
>
> - We conducted two additional experiments (noisy labels and partially available labels), available here: [Link](https://anonymous.4open.science/r/figsstuff-7F69/noise.jpg).
>   - These results show that **COBRA remains effective** and continues to reduce bias **even under substantial label noise** and limited group supervision.
>   - **Notably,** even with only partial group labels, COBRA still consistently improves fairness over Vanilla DD.
>   - **Reason:** COBRA enforces alignment across subgroup representations, reducing reliance on subgroup-specific shortcuts and encouraging the model to learn **more stable and transferable features**, which improves robustness under label noise and limited group supervision.
>   - **Handling missing group labels:** Labels are estimated using a simple unsupervised procedure (K-means clustering followed by cluster-wise majority-vote pseudo-labeling), enabling COBRA to operate under limited supervision.

---

> > ### Author Rebuttal · Reviewer_pF7z · 2026-03-31
> >
> > I've read the full rebuttal and am revising to 5 (Accept).
> >
> > The rebuttal addressed all four of my weaknesses with actual experimental results, not promises:
> >
> > - Table 2 variance (W1). Full table with standard deviations provided, confirming the 10-run protocol across all sweep settings.
> > - MTT backbone (W2). MTT experiments run and reported. The adaptation requires demographic-specific replay buffer averaging rather than a drop-in target substitution. The plug-in framing in the abstract deserves a light qualification for trajectory-based methods.
> > - GAP construction (W3). The semantic-preserving GAP experiment with orthogonal offsets and a stable FULL baseline cleanly isolates distillation-specific amplification from task difficulty.
> > - Condition (9) tightness (W4). Concrete worst-case residual numbers in violation cases show the sufficient condition is not binding in practice. These should appear in an appendix.
> >
> > One note on Table 2: at SKEW=0.60, DM, the COBRA/FairDD margin (0.46 EOD) falls within overlapping standard deviations. This is the least challenging setting in the sweep and the convergence is expected, i.e., when imbalance is mild the barycenter and vanilla mixture are similar. The sweep should be read directionally; the strong separation at higher skew (SKEW=0.85, DM: 13.27 vs. 24.84) is where the mechanism does real work. A one-sentence caveat in the paper would suffice.
> >
> > Before camera-ready, the authors ought to please qualify the plug-in framing for MTT, add the SKEW=0.60 caveat, and expand the limitations section to acknowledge EOD-only evaluation, binary/low-cardinality sensitive attribute scope, and group-label availability requirement.

---

> > > ### Author Response · Authors · 2026-03-31
> > >
> > > Thank you for your detailed and thoughtful feedback. We appreciate your evaluation and are pleased that our additional experiments and clarifications addressed your concerns. We will incorporate your suggestions in the camera-ready version.
> > >
> > > We also sincerely apologize for using this space to address points raised by other reviewers. Due to our misunderstanding of the rebuttal process, we failed to post responses in the appropriate place. We are sorry for the inconvenience and greatly appreciate your understanding.
> > >
> > > ---
> > >
> > > # Response to Reviewer **8dGH**
> > >
> > > We sincerely apologize for not posting our responses in the proper place. This was due to our misunderstanding of the rebuttal process, and it was entirely our mistake. We are sorry for the confusion and inconvenience.
> > >
> > > **W1:**
> > > - We agree that our method does not directly prove a reduction in EOD.
> > > - **To address this,** we provide an additional empirical analysis showing a clear correlation between the worst-case geometric residual and EOD: [link](https://anonymous.4open.science/r/figsstuff-7F69/residual_vs_eod.pdf).
> > > - The Spearman correlation between residual and EOD is **0.8201**, indicating a strong empirical connection.
> > > - This supports that controlling the geometric residual is closely associated with improved downstream fairness.
> > >
> > >
> > >
> > >
> > >
> > >
> > > **W3:**
> > > - We have added **additional subgroup-aware baselines at the representation level** available here:
> > > [Link](https://anonymous.4open.science/r/figsstuff-7F69/fairextract.jpg).
> > > - Specifically, we consider **LW** and **LfF** as fairness baselines by first training a fair feature extractor following prior work (Nam et al., 2020), and then applying vanilla DD on top of that extractor.
> > > - These results suggest that **bias is introduced during the distillation process itself**, rather than arising only from representation learning. In other words, **using a fair feature extractor alone is not sufficient** to ensure fair condensed data.
> > >
> > > **References:** Nam, et al. *Learning from Failure: De-biasing Classifier from Biased Classifier*. NeurIPS 2020.
> > >
> > > **W4, Q2:**
> > >
> > > - We first note that the different barycenter forms reported in **Table 4 (Sec. 5.5)** all reduce bias compared to vanilla DD, not only the quadratic form used in COBRA.
> > > - To **isolate the effect of the barycenter** itself from that of uniform weighting, we also computed a **weighted barycenter target**, where subgroup weights are **proportional to group size**.
> > >
> > > | Dataset | Method | Acc | EOD |
> > > |---|---|---:|---:|
> > > | BFFHQ | Vanilla DD | 65.1 | 44.8 |
> > > | BFFHQ | Weighted barycenter | 65.48 | 33.9 |
> > > | BFFHQ | Uniform barycenter | 69.5 | 15.7 |
> > > | CIFAR10-S | Vanilla DD | 37.2 | 56.3 |
> > > | CIFAR10-S | Weighted barycenter | 39.2 | 31.5 |
> > > | CIFAR10-S | Uniform barycenter | 44.5 | 20.2 |
> > > - These results show that **the barycenter itself already reduces bias**, even with non-uniform weights.
> > > - **Uniform weighting further improves fairness and accuracy**, which is why COBRA performs best.
> > > ----
> > >
> > > **Q1:**  We kindly refer to **W4 of Reviewer pF7z** for this question.
> > >
> > > **W2, Q3:**  We kindly refer to **W2 of Reviewer ZMwo** for this question.
> > >
> > > **Q4:** We kindly refer to **W3 of Reviewer ZMwo** for this question.
> > >
> > > ---
> > > # Response to Reviewer **GV5Z**
> > >
> > > We sincerely apologize for not posting our responses in the proper place. This was due to our misunderstanding of the rebuttal process, and it was entirely our mistake. We are sorry for the confusion and inconvenience.
> > >
> > > **W1, Q3, Limitations:**
> > > - **The choice of distance metric matters because it changes the geometry of the representation space**, and therefore changes the barycenter target used for alignment.
> > > - **Different datasets favor different geometries**, so no single metric is best in every case.
> > > - **Importantly, all tested barycentric variants still substantially outperform Vanilla DD**, so the main benefit comes from **barycentric alignment itself**, not from one specific metric choice.
> > > - **Metric tuning can provide additional gains**, but it requires solving more expensive inner optimization problems.
> > > - For this reason, we use the default COBRA metric because it **preserves the main fairness improvement while remaining simple, stable, and efficient**.
> > >
> > >
> > > **Q2:**
> > > - We added an extra experiment: [Link](https://anonymous.4open.science/r/figsstuff-7F69/Colored_MNIST_foreground_DM.png).
> > > - COBRA's fairness improvement **does not come from degrading majority-group performance**; it is driven by **improving the minority / worst-performing groups**, while the strongest groups remain largely stable.
> > > - **Reason** is that COBRA encourages the distilled data to preserve **more representative and transferable features**, improving weaker groups without materially harming stronger ones.
> > > - The additional experiment **further confirms this**, as instead of reducing the majority-group TPR, COBRA raises the minority-group TPR to close the gap.
> > >
> > > **W2, Q1, Limitations:**
> > > We kindly refer to **W2 of Reviewer ZMwo** for this question.

---

### Official Review · Reviewer_GV5Z · 2026-03-02

**Soundness:** 3
**Presentation:** 3
**Significance:** 3
**Originality:** 3
**Overall Recommendation:** 4
**Confidence:** 3

**Summary:**

This work addresses fairness in dataset distillation. The authors demonstrate that bias in distilled datasets arises from the interaction between group imbalance and differences in group representations, arguing that it is their combined effect that causes unfairness.
In this context, they introduce COBRA, a method that computes a barycenter to assign equal importance to all groups regardless of their size. Then, distillation is performed to match this barycenter, preventing any single group from dominating the final representation.
Experimental results show that COBRA reduces the EOD gap between groups, outperforming both standard distillation methods and a prior fairness-aware approach. The method is evaluated across multiple distillation backbones and datasets.

**Compliance With Llm Reviewing Policy:**

Affirmed.

**Final Justification:**

As no rebuttal was provided, my evaluation remains unchanged

**Key Questions For Authors:**

1. How sensitive is the method to noisy or incomplete group labels?
2. Does improving fairness come at a cost to the majority group?
3. Why does the choice of distance metric matter so much?

**Limitations:**

This method requires access to group labels for the entire training set. In addition, the choice of distance metric for the barycenter introduces a dataset-specific hyperparameter that may need tuning. For applicability, it would be beneficial to provide a practical criteria or a guideline for choosing the distance metric.

**Strengths And Weaknesses:**

Strengths:
- The analysis is well-grounded, moving from diagnosing why bias is amplified during distillation, and then proposing a solution to address it.
- The method works and this is evidenced by the results across multiple datasets and distillation techniques. COBRA consistently reduces the gap between groups compared to baselines.
- These gains also hold when using different architectures on the distilled data, which shows that the method is not model-specific.

Weaknesses:
- Section 5.5 shows that the best metric varies, which means some tuning may be needed per dataset.
- The method needs group labels, which limits applicability when sensitive attributes are missing or noisy.

---

### Official Review · Reviewer_ZMwo · 2026-03-08

**Soundness:** 3
**Presentation:** 3
**Significance:** 3
**Originality:** 2
**Overall Recommendation:** 4
**Confidence:** 5

**Summary:**

The authors address the interaction between dataset distillation and subgroup fairness. Dataset distillation aims to compress a large dataset into a small synthetic dataset while preserving predictive performance, but the paper argues that existing distillation objectives focus mainly on average accuracy and may amplify bias across demographic groups.
Specifically, the paper argues that subgroup disparities emerge not solely from group imbalance or differences in subgroup representations, but from the interaction between these two factors. The authors formalize this phenomenon and derive an upper bound characterizing how subgroup imbalance and representational separation jointly influence the residual mismatch between subgroup statistics and the distilled representation.

**Compliance With Llm Reviewing Policy:**

Affirmed.

**Final Justification:**

Based on the detailed theoretical proof and reliable experimental results provided by the author, I ultimately recommend weak accept and did not give a higher score because the research direction of fair dataset distillation is somewhat narrow

**Key Questions For Authors:**

Please refer to the weaknesses.

**Limitations:**

Yes

**Strengths And Weaknesses:**

Strengths

- The paper tackles fairness issues in dataset distillation, an area that has received relatively limited attention despite the increasing use of distilled datasets in practical settings.

- The paper provides a formal derivation showing how the class-conditional distilled representation is pulled toward the mixture of subgroup statistics, which can lead to subgroup-specific residual errors when subgroup representations are misaligned.

- The proposed COBRA framework modifies the target representation used in distillation by replacing the mixture representation with a group-balanced barycenter. This design is conceptually simple and can be integrated into multiple distillation objectives without modifying downstream training procedures.

Weaknesses

- The core idea of computing a barycenter across subgroup representations is conceptually related to prior fairness methods that use optimal transport or barycentric aggregation to mitigate bias. While the paper adapts this idea to dataset distillation, the methodological novelty may be viewed as incremental.

- The proposed method requires access to protected group labels in order to compute subgroup statistics and barycentric targets. In many real-world settings such labels may be unavailable, incomplete, or sensitive, which could limit the applicability of the method.

- The COBRA objective introduces an additional step of computing subgroup barycenters during distillation. While the authors note that closed-form solutions exist for certain discrepancy choices, the additional optimization may increase computational complexity compared with standard distillation objectives.

---

> ### Author Rebuttal · Authors · 2026-03-31
>
> **W1:**
>
> We thank the reviewer for the feedback. We agree that optimal transport and barycentric aggregation are established tools. Our main contribution is **not** merely applying them in a new setting, but **identifying and formalizing the source of bias in dataset distillation**. Specifically, we show that distillation amplifies **conflicting subgroup patterns** and **skewed group ratios**, which together worsen bias after distillation. This analysis naturally motivates COBRA, making the barycenter construction a **principled solution** rather than an ad hoc design.
>
> -------
> **W2:**
>
> - We conducted two additional experiments (noisy labels and partially available labels), available here: [Link](https://anonymous.4open.science/r/figsstuff-7F69/noise.jpg).
> - These results show that **COBRA remains effective**, continuing to reduce bias **even under substantial label noise** and limited group supervision.
> - **Notably,** even with only partial group labels, COBRA still consistently improves fairness over Vanilla DD.
> - **Reason:** COBRA enforces alignment across subgroup representations, reducing reliance on subgroup-specific shortcuts and encouraging the model to learn **more stable and transferable features**, which improves robustness under label noise and limited group supervision.
> - **Handling missing group labels:** Labels are estimated using a simple unsupervised procedure (K-means clustering followed by cluster-wise majority-vote pseudo-labeling), enabling COBRA to operate under limited supervision.
>
> -------
> **W3:**
> - We added **computational overhead analysis** (time and memory), available here: [Link](https://anonymous.4open.science/r/figsstuff-7F69/Time.jpg).
>   - **Theory:** COBRA introduces **no additional inner optimization loop**. The subgroup barycenter has a **closed-form** solution with cost $\mathcal{O}(Gd)$, where $G$ is the number of demographic groups (typically $<10$) and $d$ is the feature dimension.
>   - **Practice:** The main overhead comes from replacing one class-level real-data pass with $G$ group-wise passes, and the measured overhead remains **modest and predictable** in practice.

---

> > ### Author Rebuttal · Reviewer_ZMwo · 2026-04-01
> >
> > My concerns are fully resolved now with the additional provided clarification, and i will keep my positive score

---

> > > ### Author Response · Authors · 2026-04-02
> > >
> > > Thank you for taking the time to read our rebuttal. We appreciate your positive assessment and are glad the additional clarification helped address your concerns.

---

### Official Review · Reviewer_8dGH · 2026-03-08

**Soundness:** 3
**Presentation:** 3
**Significance:** 3
**Originality:** 3
**Overall Recommendation:** 5
**Confidence:** 5

**Summary:**

This paper studies fairness in dataset distillation. The paper starts from the observation that standard distillation matches a single aggregate target representation, which can overfit majority-group structure when demographic groups differ in both size and predictive patterns. The authors analyze this failure mode through the interaction between group imbalance and subgroup representational separation, derive a residual-based bound, and propose COBRA, a simple fairness-aware objective that replaces the vanilla class-conditional target with a cross-group barycenter. The method is designed to be plug-in compatible with existing distillation backbones. The experiments cover several biased synthetic and real-world benchmarks and show large reductions in equalized-odds difference, often with improved accuracy as well.

**Compliance With Llm Reviewing Policy:**

Affirmed.

**Final Justification:**

After reading the rebuttal and considering the other reviewers’ comments, I find that my concerns have been well addressed. The authors provided satisfactory clarifications, and I will therefore increase my score to Accept.

**Key Questions For Authors:**

1. Theorem 4.1 is a central part of the paper’s justification. Can the authors clarify how often the directional condition in Eq. (9) is expected to hold in practice, and whether there is any empirical evidence that the theorem’s condition is usually satisfied on the reported benchmarks?

2. The practical COBRA implementation reduces the barycenter to a uniform mean under the chosen discrepancy. To what extent is the observed gain coming from the barycentric view itself, as opposed to simply replacing a skewed aggregate target with a uniform subgroup average in representation space?

3. The method assumes access to protected-group annotations during distillation. How sensitive is COBRA to noisy or incomplete group labels, and do the authors expect the method to remain useful when only partial subgroup information is available?

4. Could the authors comment on the additional computational cost of COBRA relative to the underlying DD objective?

**Limitations:**

Yes

**Strengths And Weaknesses:**

Strengths:
1. The paper addresses an important topic. Fairness in dataset distillation is still underexplored, and the paper identifies a real issue: distilled datasets can amplify subgroup disparities even when average performance looks acceptable. The study's major contribution concerns a clean conceptual reframing of fair distillation as cross-group barycenter alignment in representation space, rather than only loss reweighting or rebalancing. I find this framing intuitive and useful. The method itself is also simple enough to be practical. COBRA only changes the target statistic and can, in principle, be combined with multiple distillation methods, which is an attractive design choice.

2. The empirical results are strong overall. On the biased benchmarks in Table 1, COBRA consistently improves EOD and often improves accuracy at the same time. The gains are especially striking on C-MNIST, C-FMNIST, CIFAR10-S, and BFFHQ. The paper also includes controlled experiments that separately vary skew and representation gap, which helps support the core narrative that both factors matter. In addition, the cross-architecture evaluation is a useful check, since fairness gains that only hold for one evaluation model would be much less convincing.

3. I also appreciate that the paper makes a real effort to connect the method to theory. The residual analysis in Section 4 gives a plausible explanation of how vanilla distillation can produce subgroup-specific mismatch, and Theorem 4.1 gives at least a partial guarantee that the COBRA target does not worsen the worst-case residual under the stated condition. This is not a deep theory paper, but the theoretical component is aligned with the method rather than being decorative.

Weaknesses:
1. The main theorem depends on an additional directional condition, and the practical meaning of that condition is not fully clarified. The paper argues that COBRA reduces the worst-case residual under this setting, but the theory stops short of giving a direct fairness guarantee on the downstream predictor. As a result, the bridge from residual geometry to EOD remains suggestive rather than fully formalized.

2. This is reasonable for the problem studied here, but it should be stated more explicitly as a scope limitation. In many practical settings, protected attributes may be missing, noisy, legally restricted, or only partially available. The current method does not address that case, and the applicability of the approach depends strongly on this assumption. The paper briefly mentions this in the conclusion as future work, but it deserves more explicit discussion.

3. The paper compares mainly against vanilla DD, FairDD, and several standard distillation backbones. This is a reasonable starting point, but the fairness-aware comparison set is still narrow. Since the paper positions itself as a fairness-oriented distillation method, it would be stronger to compare more directly with other subgroup-aware preprocessing or representation-level fairness baselines, even if they are not designed specifically for DD.

4. The practical implementation specializes the barycenter discrepancy to a quadratic form so that the inner problem reduces to a uniform mean over subgroup statistics. This makes the method efficient, but it also weakens the impression that the barycenter machinery itself is essential. In fact, the ablation suggests that alternative discrepancies behave quite inconsistently across datasets. This makes COBRA look partly like a well-motivated uniform subgroup averaging scheme in representation space, rather than a fully robust barycentric principle.

---

### Decision · Program_Chairs · 2026-04-30

**Decision:**

Accept (regular)

**Comment:**

Overall, the reviews are consistently positive, with two accepts and two weak accepts. The reviewers raised several concerns, which have been well addressed in the rebuttal.

Hence, I recommend accepting the paper.